# Distinct mechanisms mediate X chromosome dosage compensation in *Anopheles* and *Drosophila*

Claudia Isabelle Keller Valsecchi[1,3,]*, Eric Marois[2,]*, M Felicia Basilicata[1,3,]*, Plamen Georgiev[1], Asifa Akhtar[1]

Sex chromosomes induce potentially deleterious gene expression imbalances that are frequently corrected by dosage compensation (DC). Three distinct molecular strategies to achieve DC have been previously described in nematodes, fruit flies, and mammals. Is this a consequence of distinct genomes, functional or ecological constraints, or random initial commitment to an evolutionary trajectory? Here, we study DC in the malaria mosquito *Anopheles gambiae*. The *Anopheles* and *Drosophila* X chromosomes evolved independently but share a high degree of homology. We find that *Anopheles* achieves DC by a mechanism distinct from the *Drosophila* MSL complex–histone H4 lysine 16 acetylation pathway. CRISPR knockout of *Anopheles msl-2* leads to embryonic lethality in both sexes. Transcriptome analyses indicate that this phenotype is not a consequence of defective X chromosome DC. By immunofluorescence and ChIP, H4K16ac does not preferentially enrich on the male X. Instead, the mosquito MSL pathway regulates conserved developmental genes. We conclude that a novel mechanism confers X chromosome up-regulation in *Anopheles*. Our findings highlight the pluralism of gene-dosage buffering mechanisms even under similar genomic and functional constraints.

## Introduction

Chromosome-wide copy number alterations are typically linked to developmental failure and diseases such as cancer. Heteromorphic sex chromosomes are a notable exception. To account for potentially deleterious expression imbalances in XY males in comparison with XX females, X chromosomes are often subjected to dosage compensation (DC) (Samata & Akhtar, 2018). In Dipterans such as fruit flies or mosquitos, sex chromosomes have formed multiple times (Vicoso & Bachtrog, 2015) and show high evolutionary turnover (Herpin & Schartl, 2015). RNA-seq studies revealed that X chromosome DC in multiple Dipteran genera occurs by up-regulation of the single male X (Rose et al, 2016). Owing to their short generation times and related, yet rapidly evolving genomes, Dipteran species are excellent models to understand how cells co-opt pathways to keep chromosome-wide gene dosage alterations in check. Dipterans have also been intensively studied as some of its species are prominent disease vectors, for example, the malaria-transmitting mosquito *Anopheles gambiae*. Understanding the processes responsible for sex differentiation and regulation of heteromorphic sex chromosomes is highly relevant as these factors are among the most promising targets for gene-drive strategies aiming at preventing transmission of infectious diseases by vector control (Kyrou et al, 2018).

DC in the fruit fly *Drosophila melanogaster* (Dmel) is mediated by association of the male-specific lethal (MSL) complex with the X chromosome in males. Via deposition of histone H4 lysine 16 acetylation (H4K16ac), the MSL complex enhances transcription to achieve approximately twofold up-regulated expression of X-linked genes (Lucchesi & Kuroda, 2015; Samata & Akhtar, 2018). Loss of MSL complex member genes results in male-specific lethality, whereas females remain unaffected (Belote & Lucchesi, 1980). Key to establishing MSL-DC is its subunit MSL2, which is male-specifically expressed and recruits the complex to the X chromosome (Bashaw & Baker, 1995). H4K16ac enrichment on the X as well as other characteristic features of the MSL-DC system are conserved in other drosophilids, for example, the distantly related *Drosophila virilis* or *Drosophila busckii* (Alekseyenko et al, 2013; Renschler et al, 2019). The MSL complex is also present in mammals, but its in vivo functions remain poorly characterized and mammalian DC is mediated by X chromosome inactivation in females via an entirely different pathway (Keller & Akhtar, 2015).

Like *Drosophila*, the malaria mosquito *A. gambiae* (Agam) achieves DC by hypertranscription of the male X (Deitz et al, 2018). The Dmel and Agam X chromosomes evolved independently from the same ancestral autosome (Zdobnov et al, 2002). It remains unclear whether X up-regulation in mosquitos is achieved by the same molecular machinery as in drosophilids or whether different mechanisms may arise in these contexts. This is of particular interest, as similar functional constraints and gene contents have shaped the acquisition of DC on newly evolving sex chromosomes in these lineages. Hence, comparing Agam with Dmel offers an excellent opportunity to study whether the independent evolution of sex chromosomes is accompanied by the acquisition of the same DC mechanism.

[1]Max Planck Institute of Immunobiology and Epigenetics, Freiburg, Germany  [2]Université de Strasbourg, Centre National de la Recherche Scientifique (CNRS) UPR9022, Inserm U1257, Strasbourg, France  [3]Institute of Molecular Biology (IMB), Mainz, Germany

Correspondence: akhtar@ie-freiburg.mpg.de; c.keller@imb-mainz.de
*Claudia Isabelle Keller Valsecchi, Eric Marois, and M Felicia Basilicata contributed equally to this work

Here, we analyzed whether the MSL complex–H4K16ac pathway confers X chromosome–wide DC in Agam. Expression analyses indicate that unlike in *Drosophila*, the MSL complex subunits are expressed in both males and females in *Anopheles*. Loss of *Anopheles msl-2* affects both sexes, where zygotic KO induces an early embryonic lethal phenotype and germ line KO is results in sterility. Transcriptome analyses indicate that *msl-2* KO embryos do not display a global perturbation of X-linked gene expression. Furthermore, ChIP-seq and immunofluorescence show that H4K16ac is neither enriched on the male mosquito X chromosome nor displays a sexually dimorphic pattern. Instead, the MSL–H4K16 pathway appears to be involved in regulating developmental genes. Our study highlights that X chromosome DC can be achieved by entirely different molecular strategies even in very similar genomic contexts.

# Results

## MSL complex members are expressed in both male and female *A. gambiae*

Cytogenetic and genomic studies suggest that the ancestral dipteran karyotype consisted of six chromosomal pairs commonly referred to as Muller elements A-F (Vicoso & Bachtrog, 2015). Two independent evolutionary events resulted in Muller element A becoming sex-linked in *Drosophila* and *Anopheles* (Fig 1A) (Landeen & Presgraves, 2013; Vicoso & Bachtrog, 2015). The X chromosomes of these species exhibit considerable homology (Fig 1B [Zdobnov et al, 2002]), with around half of the Agam X-linked genes sharing an annotated X-linked orthologue in Dmel (Fig 1C).

To elucidate whether the MSL complex mediates DC in Agam, we first analyzed the amino acid sequence conservation of the core MSL subunits MSL1, MSL2, MSL3, and MOF, as well as the associated RNA helicase MLE (Table S1 and Fig 1D; note that the high evolutionary turnover of noncoding RNAs hinders the computational identification of putative orthologues of Dmel *roX1* and *roX2* [Quinn et al, 2016]). The highest degree of conservation was found for MLE (57% identity) and MOF (61% identity), the latter of which displays a shortened N-terminus in *Anopheles* compared to *Drosophila*. MSL3 displays a similar length and more than 50% similarity for both the chromo- and MRG domains. The RING and CXC domains of MSL2 are also conserved, but the flexible linker regions adjacent to these structured domains are substantially extended in *Anopheles*. The DNA binding CXC domain of Agam MSL2 appears more similar to mammalian than *Drosophila* MSL2 (Fig S1A). There is no annotated Agam *msl-1* gene and extensive BLAST searches using the full-length protein failed to identify a bona-fide orthologue of MSL1. Because MSL1 is largely unstructured (Hallacli et al, 2012), the overall sequence level conservation may be rather low. Indeed, individual domain searches revealed hits for the coiled-coil and PEHE regions, albeit at rather low fidelity (Table S1; note that even mammalian and *Drosophila* MSL1 show <20% amino acid similarity despite providing the same scaffolding function to the complex).

Assessing the expression level of the core MSL complex members by RT-qPCR from adult *Anopheles*, we found a modestly biased expression of *mof* and *mle* in males (Fig 1E, statistical analyses in

Table S1). Since entire adult mosquitos contain the gonads, which could be introducing a bias, we also analyzed published RNA-seq data (Papa et al, 2017) of dissected carcasses (soma) and reproductive tissues (germline) (Fig S1B; modEncode and RT-qPCR data from Dmel as a comparison in Fig S1C and D). In this RNA-seq dataset, Agam *msl*s were all classified as not sex-biased. This discrepancy could be a consequence of differences in sensitivity (RT-qPCR versus RNA-seq), or alternatively, other tissues with male-biased expression present in the sample analyzed by us.

Next, we turned our attention to the expression pattern of *msl*s during *Anopheles* embryogenesis (mixed-sex data from Goltsev et al [2009]). *msl-3* and *mof* mRNA levels drop along development, whereas *mle* and *msl-2* appear continuously expressed (Fig 1F). In *Drosophila*, *msl-2* and *mle* expression exhibits a different pattern during embryogenesis, with a pronounced induction shortly after zygotic genome activation (ZGA) ([Lott et al, 2011; Samata et al, 2020], Fig S1E).

In *Drosophila*, both sexes express MOF and MLE proteins (Fig 1G and H [Kotlikova et al, 2006; Conrad et al, 2012]), and at least in some tissues, MSL1 and MSL3 are not only present in males, but also in females (Chlamydas et al, 2016; McCarthy et al, 2019 *Preprint*). However, posttranscriptional mechanisms prevent MSL2 protein expression in females thereby providing the DC function specifically to males (Beckmann et al, 2005). As antibodies against Agam MSL2 are unavailable, we assessed its protein production by polysome profiling. This revealed that unlike *Drosophila*, MSL2 protein is produced in both male and female mosquitos (Figs 1I and S1F).

## *msl-2* KO in *A. gambiae* triggers sex-independent phenotypes

Our observation that unlike *Drosophila*, MSL2 is expressed in both male and female mosquitos prompted us to investigate whether it is involved in sex-specific functions or in a more general role in gene expression regulation. We generated transgenic Agam for conditional KO of the Agam *msl-2* gene by CRISPR/Cas9 (Fig 2A, see the Materials and Methods section). We crossed *vasa-Cas9* mothers with *msl-2* gRNA transgenic fathers, where Cas9 protein is maternally deposited into the fertilized egg. This results in *msl-2* KO shortly after the onset of zygotic transcription of the paternally provided gRNA transgenes (Fig 2A). Compared to the controls, which completed embryonic development within around 42 h followed by hatching (Goltsev et al, 2007), the *msl-2* KO progeny displayed an arrest in early embryogenesis (Figs 2B and S2A). To analyze cell death, we performed TUNEL stainings in embryos collected at 6, 16 and 24 h after egg laying. TUNEL-stained cells could be detected in *msl-2* KO embryos around 16 h after egg laying and this became much more pronounced after 24 h of development (Figs 2C and S2B). Compared to Dmel, where MSL2 loss results in male-specific lethality beginning to manifest at the third instar larval stage (after around 5 d of development), this severe early embryonic phenotype affecting both sexes was unexpected. However, it was in line with expression analyses by RT-qPCR (see above), where both *msl-2* and *msl-3* are highly transcribed at ZGA, which occurs around 2–4 h after egg laying (Figs 2A and S2C). RNA levels of early patterning genes such as *hunchback*, *elovl* or *eve* as well as *msl-3* or the X-linked *AGAP000634* appeared rather unaffected in *msl-2* KO Agam (see below for transcriptome-wide analyses). RT–PCR from these mixed-sex populations collected at different time points after fertilization

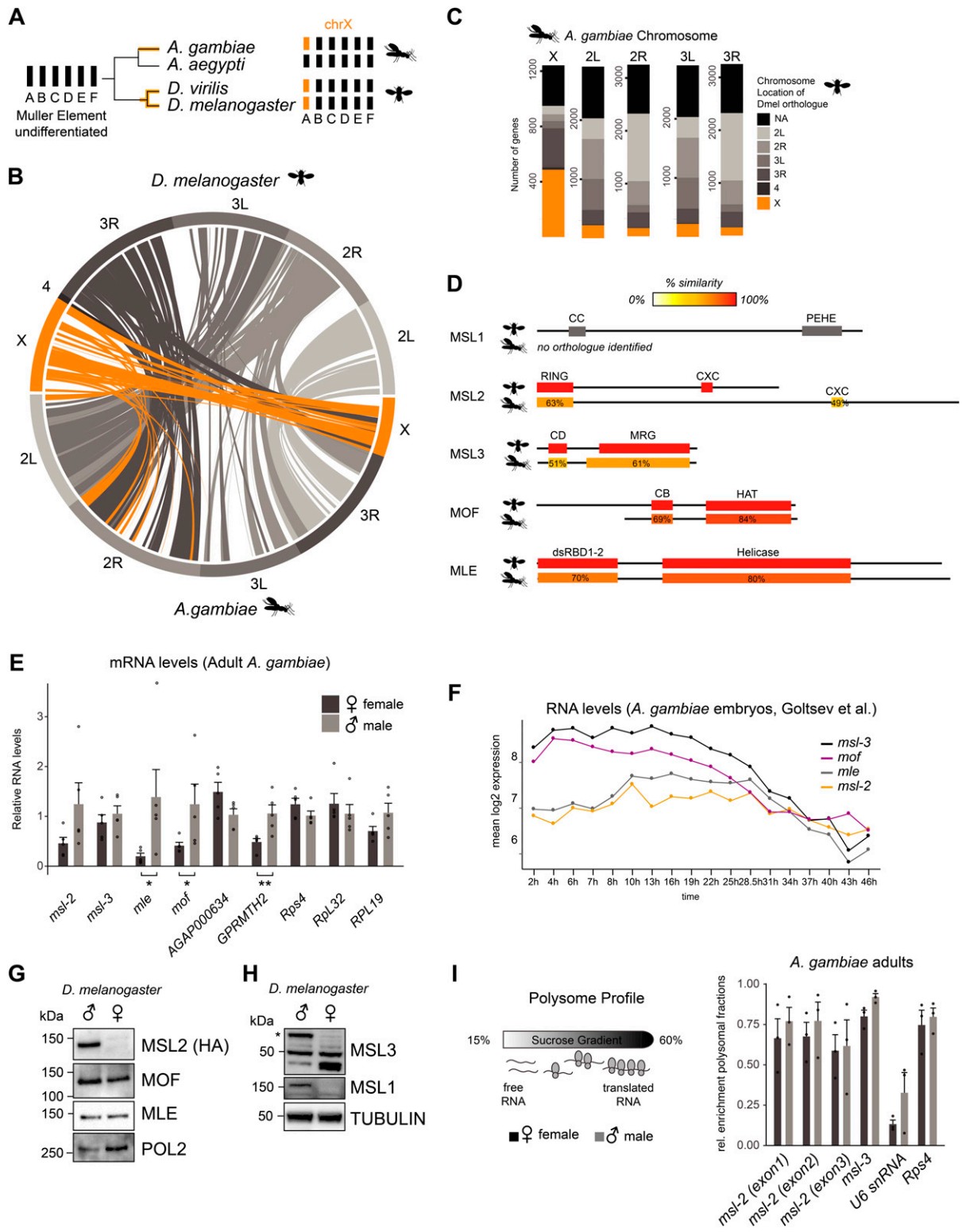

**Figure 1 MSL complex members are expressed in both male and female *Anopheles gambiae*.**
**(A)** Schematic representation of the evolution of the X chromosomes in the indicated species from the ancient Dipteran karyotype consisting of Muller elements A-F. Note that the mosquito *Aedes aegyptii* does not contain a differentiated sex chromosome. **(B)** Circos plot of pairs of synteny blocks that were generated over genomic regions of at least 100 kb among *Drosophila melanogaster* and *Anopheles gambiae* genomes. Synteny blocks on the Dmel X chromosome are shaded in orange, autosomes in grey. **(C)** Bar plot illustrating the chromosomal location of 1:1 orthologues of all *A. gambiae* genes in the *D. melanogaster* genome. If a given *A. gambiae* gene does not contain an annotated *D. melanogaster* orthologue, the bar is colored in black, X-linked genes in orange and genes on the autosomal arms in grey. Orthologues were

revealed that there were both males (assessed by expression of *Yob* [Krzywinska et al, 2016]) and females (assessed by female-spliced *Dsx* [Kyrou et al, 2018]) among the transcriptionally active population (Fig 2D). This was validated by single-embryo genotyping by qPCR, where we detected approximately equal numbers of males and females in the F1 embryos across different developmental time points (Fig 2E). Therefore, the proportion between the two sexes is unchanged by *msl-2* KO throughout embryogenesis until their arrest, indicating that both sexes are equally affected upon loss of MSL2.

In *Drosophila*, the loss of *msl-2* is linked to a bulk decrease of its downstream epigenetic mark, H4K16ac. When comparing *Anopheles* wild-type and *msl-2* KO embryo extracts by Western blot, we found no scorable differences in H4K16ac levels (Fig 2F). In summary, these findings point towards an essential function for MSL2 in both male and female mosquitos early in embryogenesis that is distinct from *Drosophila*.

We next crossed *msl-2* gRNA-expressing mothers with *vasa-Cas9* transgenic fathers, which induces a KO in germ line tissues of the F1 progeny. When F1 males were crossed to wild-type females, normal numbers of eggs were laid but those did not hatch, indicating that the F1 males were sterile (pictures of testis in Fig S2D). F1 females displayed a drastic atrophy of the ovaries and failed to lay eggs even after two consecutive blood meals, whereas in the control, a single blood feeding reliably triggered ovary development and egg laying (Fig 2G). This phenotype is again distinct from *Drosophila*, where *msl-2* KO females are fertile (Belote & Lucchesi, 1980). We performed immunofluorescence stainings of H4K16ac in female F1 KO ovaries (Fig 2H; note the different scale bars and overall differences in organ size in control compared with KO). A range in severity of phenotypes was observable. Most females displayed a complete absence of the germ line tissue, where only the wild-type somatic ovary epithelium was present. Some individuals exhibited a defective ovarian tissue that still contained an oocyte. Overall, the tissue architecture and morphology were drastically compromised upon *msl-2* KO in the female germ line (Videos 1 and 2). We conclude that MSL2 is essential for faithful progression of embryonic and germ line development in both male and female mosquitos.

### MSL2 confers gene-by-gene regulation on all chromosomes

The early embryonic and pronounced germ line *msl-2* KO phenotypes could be a consequence of an additional function of the MSL complex in mosquitos compared with *Drosophila*, which could mask a phenotype related to regulating the *Anopheles* X chromosome. To analyze this, we characterized the transcriptome alterations upon loss of *msl-2* in *Anopheles* embryos, in particular with regards to the chromosomal location. We analyzed two different time points: 6–7 h after egg laying with *n* = 3 biological replicates and 15 h after egg laying with *n* = 2 biological replicates (Fig S3A; we also generated a single dataset at 24 h after egg laying, which was only considered for qualitative analyses). Because it is not possible to phenotypically separate sexes at this stage, we performed RNA-seq experiments from mixed embryo populations. To validate that this approach per se permits capturing effects on DC of the X, we performed an equivalent experiment from a mixed male and female Dmel embryo population (Fig 3A). We scored differentially expressed (DE) genes using DESeq2 with a false-discovery rate (FDR) cutoff of <0.05 in control versus *msl-2* KO (Fig S3B and Tables S2 and S3). As expected from the role of Dmel MSL2 in regulating the X chromosome, we made the following observations in the *Drosophila* dataset: First, X-linked genes are significantly overrepresented among all down-regulated genes in the *msl-2* KO embryos (Fig 3B, 717 of 1,304 genes, Fisher's exact test $P = 1.65645 \times 10^{-273}$). Second, genes on the X chromosome are globally down-regulated, irrespective of whether a gene is scored as DE or not (Fig 3C, statistical analyses in Table S1). Furthermore, autosomes are moderately up-regulated. Third, the fold change/extent of the down-regulation occurs in a range that is consistent with an approximately twofold, DC-like effect (Fig 3A; $\log_2$FC of −0.415 assuming a 50:50 male:female population, where males are affected by twofold and females are unaffected; grey bar). Indeed, the majority of the downregulated genes (1,103 of 1,304) are reduced to 25–75% of control levels. We conclude that X-linked perturbations due to defective DC can be reliably scored from such a mixed-population experiment.

For *Anopheles*, we first inspected the earlier dataset around 6–7 h after egg laying, and found few gene expression changes in *msl-2* KO embryos (Fig S3C and D). A total of five significantly down-regulated genes (FDR < 0.05) included *msl-2* itself, three transcripts directly adjacent to *msl-2*, and the orthologue of Dmel *mira*, an actin/myosin binding protein. These limited changes suggest that the aforementioned phenotypes (Fig 2) likely arise from loss of *msl-2*, rather than from Cas9 off-target effects. Because ZGA occurs after around 3 h of development (Goltsev et al, 2007), this also implies that ZGA and early embryo development occurs faithfully without *msl-2*. We turned our attention to the 15 h after egg laying time point, where we found 566 down-regulated transcripts, as well as 360 up-regulated transcripts (Fig 3D). Distinct from *Drosophila* (orange bars in Fig 3B, Fisher's exact test in Dmel $P = 1.65645 \times 10^{-273}$, see above), the fraction of X-linked, down-regulated genes in *Anopheles* tends to only reflect the overall

obtained from VectorBase. **(D)** Amino acid sequence conservation of the individual domains of *D. melanogaster* MSL proteins obtained by BLASTP in *A. gambiae*. Detailed results are reported in Table S1. **(E)** RT-qPCR analysis of polyA+ RNA levels of the indicated genes in adult *A. gambiae* females and males. The mRNA level of each gene was normalized to *Rps4* and expressed relative to the expression level in males. The bar plot represents the mean ± SEM with overlaid data points reflecting one biological replicate. *Rps4*, *RpL32*, and *RPL19* are control genes, *AGAP000634* and *GPRMTH2* are representative X-linked and autosomal genes, respectively. Significance was evaluated by a Wilcoxon rank sum test, *P*-value < 0.05, **P*-value < 0.01, details in Table S1. **(F)** Line plot representing the mean $\log_2$ expression levels of *msl* genes during *A. gambiae* embryogenesis. The data from Goltsev et al (2009) was obtained from VectorBase. **(G)** Cropped immunoblots of male and female extracts of *msl-2::HA* (endogenously tagged) *D. melanogaster* heads. RNA Polymerase 2 (POL2) serves as loading control. **(H)**, as in (G), Tubulin serves as loading control. The asterisk indicates the major isoform of MSL3 that is only expressed in males, the band detected at 50 kD is unspecific. The band below 50 kD accumulates only in females. **(I)** Left: schematic representation of the polysome profiling strategy to assess the protein expression of *msl*s. Extracts are fractionated on a sucrose gradient to separate mRNA that associates with actively translating ribosomes from free/not-translated RNA. Right: bar plot displaying the relative enrichment of *msl*s and controls (not translated: *U6* snRNA; translated: *Rps4* mRNA) in the polysomal fractions. The data for all fractions individually is reported in Fig S1F. The barplot represents the mean ± SEM with overlaid data points reflecting one polysome profiling experiment/biological replicate.

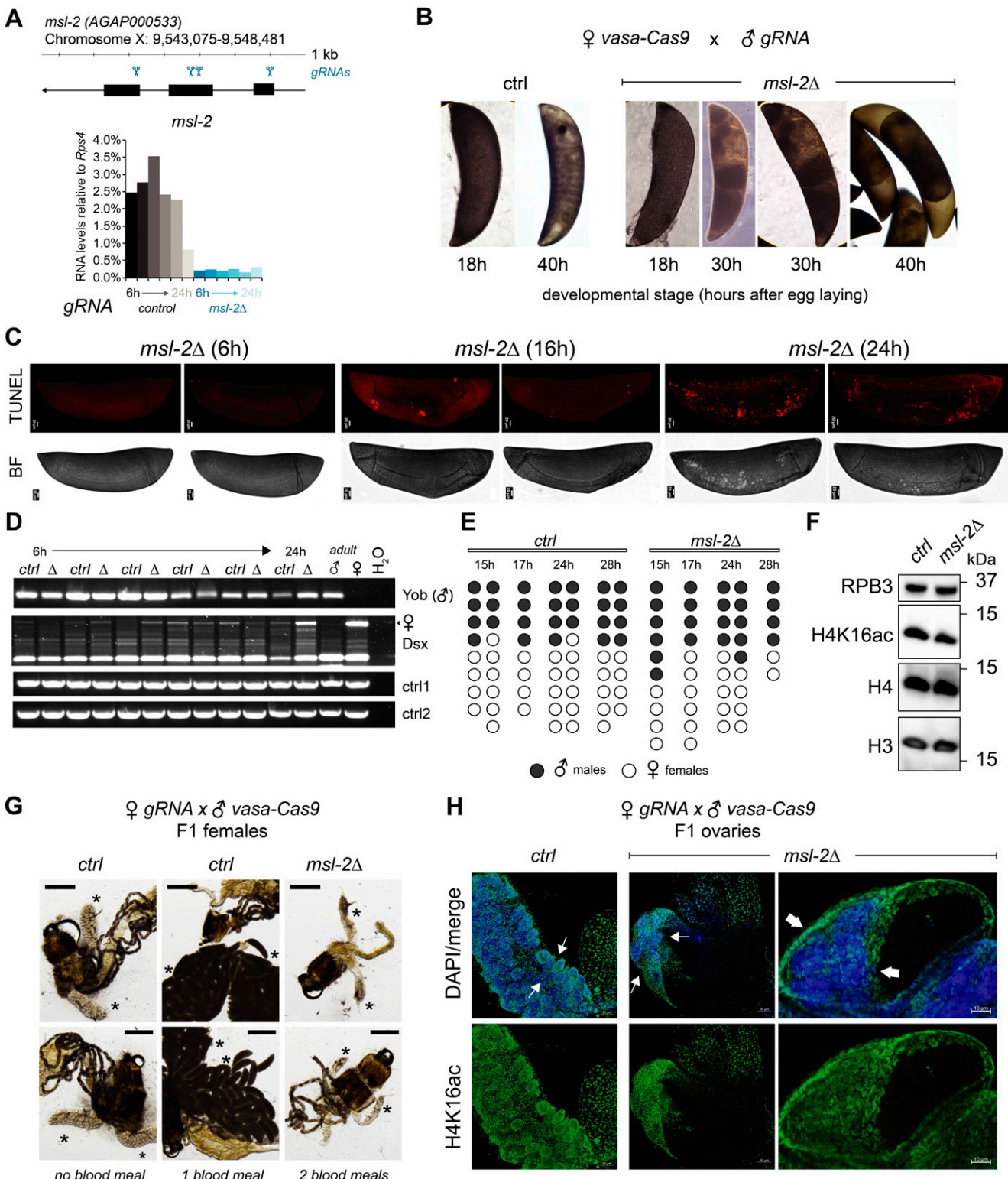

**Figure 2** *msl-2 knockout in *Anopheles gambiae* triggers sex-independent phenotypes distinct from *Drosophila*.*
**(A)** Top: schematic representation of the *A. gambiae* msl-2 gene and position of the four gRNAs that were used to generate gRNA-expressing transgenic mosquitos. Bottom: RT-qPCR analysis of msl-2 RNA levels in F1 progeny of *vasa-Cas9* mothers crossed with control (left, grey bars) or *msl-2* gRNA (right, blue bars) transgenic fathers. Each bar represents the progeny of one cross collected at the indicated time points after egg laying. **(B)** as in (A), representative pictures of control or *msl-2* gRNA-expressing transgenic F1 progeny embryos collected at the indicated time points after egg laying. The pictures were obtained with a 40× magnification on an inverted microscope. **(C)** as in (A), representative TUNEL staining and corresponding bright field (BF) images of *msl-2* gRNA-expressing transgenic F1 progeny embryos collected at

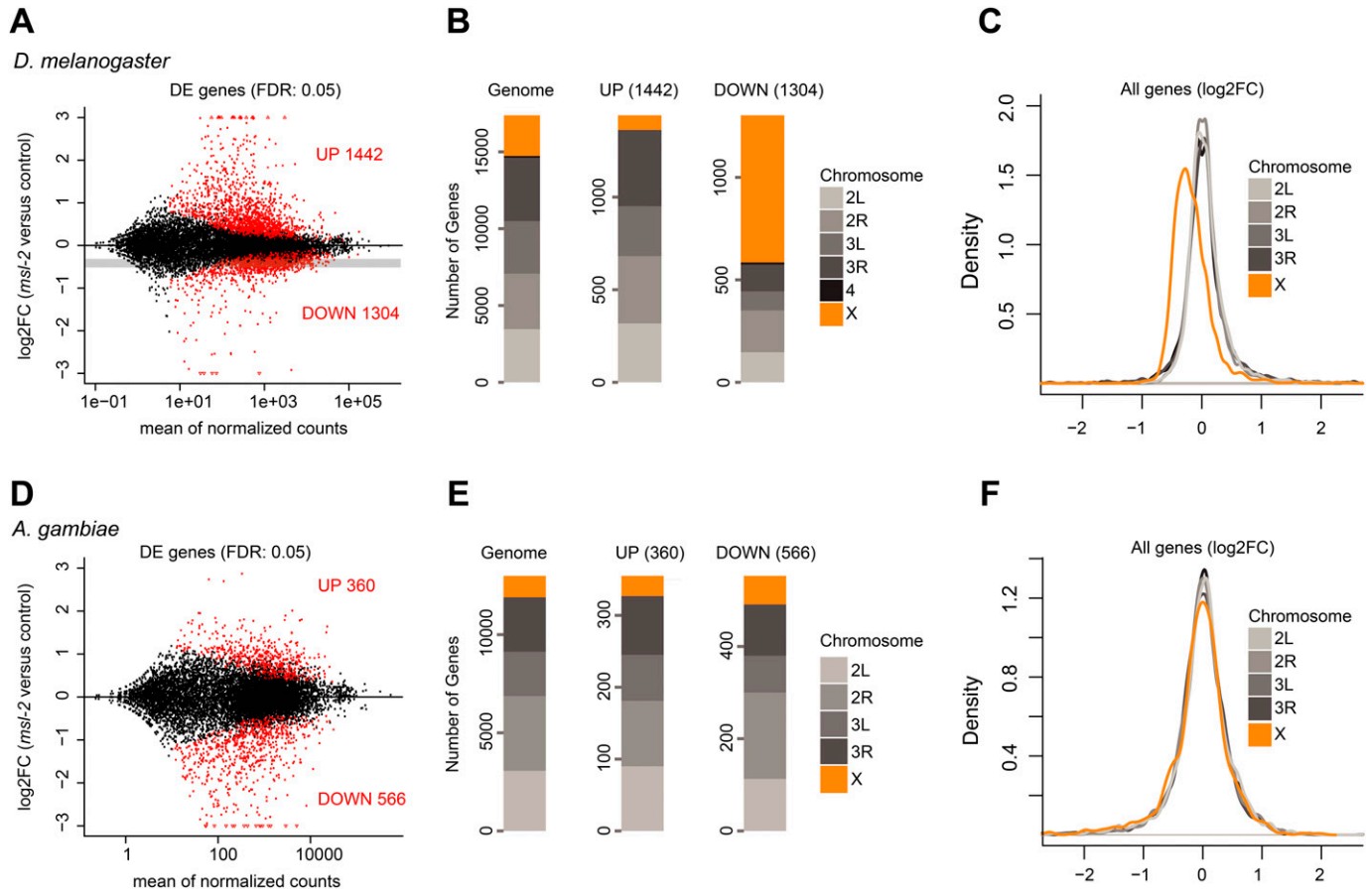

**Figure 3. Transcriptome alterations upon loss of *msl-2* in *Drosophila melanogaster* and *Anopheles gambiae* embryos.**
**(A)** RNA-seq experiments were conducted on *Drosophila melanogaster* F1 embryos obtained from crosses of *msl-2*$^\Delta$/*CyO-GFP* fathers with *msl-2*$^\Delta$/*CyO-GFP* mothers. Non-fluorescent *msl-2*$^\Delta$/*msl-2*$^\Delta$ null mutant embryos (abbreviated as *msl-2*) were compared with fluorescent *msl-2*$^\Delta$/*CyO-GFP* heterozygous controls. MA-Plots showing normalized counts versus the log$_2$(fold change) obtained with DESeq2. Differentially expressed (DE) genes (false-discovery rate < 0.05) are colored in red and the number of DE genes is reported in each panel. The grey-shaded bar indicates the expected log2FC position of down-regulated genes with a dosage compensation-like effect affected in only males but not females in a 50:50 mixed-sex population. See Table S1 for mapping statistics. **(B)** RNA-seq as in (A), Bar plot for the number of DE genes with respect to their location on the chromosomal arms. **(C)** RNA-seq as in (A), Density plots of the log$_2$(fold change) obtained with DESeq2 for genes on each of the indicated chromosomal arms. All genes were taken into account for the analyses, irrespective of whether they are scored as DE or not. **(D)** RNA-seq experiments were conducted on *A. gambiae* F1 embryos collected at 15 h after egg laying obtained from crosses of *vasa-Cas9* mothers with control or *msl-2* gRNA-expressing fathers. MA-Plots showing normalized counts versus the log$_2$(fold change) obtained with DESeq2. DE genes (false-discovery rate < 0.05) are colored in red and the number of DE genes is reported in each panel. See Table S1 for mapping statistics. **(E)** RNA-seq as in (D), Bar plot for the number of DE genes with respect to their location on the chromosomal arms. **(F)**, RNA-seq as in (D), Density plots of the log$_2$(fold change) obtained with DESeq2 for genes on each of the indicated chromosomal arms. All genes were taken into account for the analyses, irrespective of whether they are scored as DE or not.

proportion of X chromosomal genes in the genome (orange bars in Fig 3E, $n$ = 62 of 566, Fisher's exact test $P$ = 0.007221). In addition, the extent of the perturbations was much more pronounced compared with *Drosophila*, as the majority of down-regulated targets (523 of 566)

were affected by at least 75% or more (Tables S2 and S3 and Fig 3D). An overall down-regulation of the X chromosome or an up-regulation of autosomes was also not detectable in the *Anopheles* dataset (Fig 3F and Table S1). We conclude that loss of MSL2 in *Anopheles* embryos

---

the indicated time points after egg laying. The control embryos did not show a staining (Fig S2B). The pictures were cropped and rotated to orient them properly according to anterior-posterior and dorsal–ventral axis (uncropped pictures in Fig S2B). Scale bar = 20 *μm*. **(D)** cropped agarose gel of semi quantitative RT–PCR products obtained from mixed-sex F1 progeny obtained from crosses set as in (A). The RT was primed with gene-specific primers for the male-specifically expressed Yob, the primers for Dsx detect the splice variant exclusively present in females (upper band, arrow) or in both sexes (lower band). *ctrl1* and *ctrl2* RNAs serve as loading controls (see primer list). **(E)** Result of individual F1 embryo genotyping (cross as in [A]) as assessed by the expression of male-specific *Yob* by RT-qPCR. The single embryos were collected at the indicated time points after egg laying. *Rps4* was amplified as a control expressed in both males and females (not shown in figure). **(F)** Cropped immunoblots of extracts obtained from mixed-sex F1 embryos (cross as in [A]). RPB3 serves as a loading control. **(G)** representative pictures of dissected abdominal tissues of female F1 progeny obtained from crosses of *vasa-Cas9* fathers with control (ctrl) or *msl-2* gRNA-expressing mothers. The resulting *msl-2* KO is induced in the F1 germline tissue, stars point towards the ovaries. Blood feeding triggers egg laying in ctrl, but not *msl-2* KO embryos, where ovaries are completely atrophied. Scale bar = 500 *μm*. **(H)** Immunofluorescence for H4K16ac (green) in female F1 ovaries obtained from crosses as in (G). The pictures are single z-planes with scale bar = 50 *μm* (left and middle panel) or 10 *μm* (right panel) with DAPI shown in blue. Note the gross overall size and difference in tissue morphology. The arrows highlight the grape-shaped germ line tissue (see ctrl), which are highly atrophied (middle) or completely absent (right) in the KO females.

leads to global changes in gene expression affecting all chromosomes (for functional analyses on MSL2-evoked transcriptome alterations not related to X-chromosome–wide DC see below).

### Tissue-specific distribution of histone H4 lysine 16 acetylation in *A. gambiae*

The induction of a subnuclear territory enriched for an epigenetic mark in one of the sexes (H4K16ac in male Dmel, histone H3 lysine 27 trimethylation [H3K27me3] in female mammals, and histone H4 lysine 20 monomethylation [H4K20me1] in hermaphrodite *Caenorhabditis elegans*) is a characteristic feature of all known DC systems (Wells et al, 2012). To analyze this, we performed immunostainings in adult male and female mosquitos and characterized the tissue-specific enrichment of H4K16ac, in particular with regards to the presence of a X chromosome territory (Fig 4A). H4K16ac exhibited homogeneous staining of the entire nucleus in male midguts (panel 4) and Malpighian tubules (panel 3). The testis (panels 1 and 2) showed a staining pattern where H4K16ac levels were varying across the different cellular stages of sperm differentiation. The early germ cells appeared enriched, wereas the more mature, transcriptionally inert stages (e.g., spermatocytes) were depleted of H4K16ac. This cell type–specific pattern of H4K16ac in the testis was antagonized by the presence of the repressive mark H3K27me3 (Fig S4A). In females, the midguts and Malpighian tubules displayed a similar staining pattern for H4K16ac as in males (Fig 4B, panels 2 and 3; Fig 4C and D for a comparison of Agam to the X chromosome territory in Dmel). The somatic cells of the ovaries showed a high staining intensity and the oocyte itself was H4K16ac-positive. Similar to males, H3K27me3 was homogeneously staining the female nucleus and showed a tissue-specific distribution, where H4K16ac high cells displayed low levels of H3K27me3 and vice versa (Fig S4B). However, despite extensive analyses, we could not identify any cell type in adult *Anopheles* that showed a nuclear substructure akin to a putative X chromosomal territory that is marked by H4K16ac.

### Genome-wide profiles reveal H4K16ac on active genes in mosquitos

The fact that H4K16ac is not associated with a territory in *Anopheles* could be a consequence of a less pronounced X chromosome enrichment and hence, immunofluorescence not being sensitive enough to score it. We therefore generated high-resolution MNase-ChIP-seq profiles of H4K16ac, as well as a total H3 control from adult male and female mosquitos. We segmented the genome into 2 kb bins and analyzed the Input normalized ChIP enrichment on each of the chromosomal arms (Fig 5A). Although this approach readily revealed H4K16ac enrichment on the Dmel male X chromosome (Valsecchi et al, 2018), both *Anopheles* males and females looked rather similar to *Drosophila* females. Next, we used an unsupervised clustering approach to identify different enrichment patterns at male in comparison with female transcription start sites (TSSs) (Fig 5B). In *Drosophila*, this approach revealed a Cluster 1, which displays a markedly higher and broad enrichment of H4K16ac in males, but not females. Cluster 1 consisted exclusively of X-chromosomal genes (Fig 5C). Cluster 2 shows a TSS-proximal H4K16ac

signal in both males and females. In contrast to *Drosophila*, the clustering approach did not reveal differences at TSS between male and female mosquitos (Fig 5B) and/or a predominant enrichment of X-linked genes in any of the clusters (Fig 5C). Overall, the H4K16ac enrichment on genes was highly correlated between Agam males and females, but did not show the striking enrichment as on Dmel male X-linked genes (Fig S5A and B; see Fig S5C for Dmel H4K16ac data from different tissues). In line with H4K16ac promoting transcription, the level of H4K16ac on *Anopheles* genes (low to high by ChIP enrichment in quartiles Q1-Q4) was correlating with low to high read counts at the RNA expression level in RNA-seq (Fig S5D). Agam genes that showed the highest levels of H4K16ac were involved in biological processes broadly involving cellular metabolism and provided molecular functions such as "RNA binding," "signaling receptor activity" or "molecular transducer activity" (Fig S6A). Our findings were corroborated by visual inspection of the ChIP data in the genome browser (Fig 5D and E). Compared to the H3 control, H4K16ac in *Anopheles* appeared clearly enriched on individual, active genes on both X and autosomes. This indicates that TSS-proximal H4K16ac promotes gene expression in a similar fashion in Agam males and females and thereby serving a more general role in gene regulation, instead of sex-specific functions.

### *A. gambiae* MSL2 regulates conserved developmental genes

The aforementioned observations indicate that the MSL complex-H4K16ac pathway is unlikely to mediate chromosome-wide DC in Agam. We were therefore interested in which functions and gene-regulatory networks MSL2 might be controlling. To this end, we analyzed our RNA-seq dataset for the enrichment of distinctive features among the DE genes. This revealed that DE genes in *msl-2* KO embryos tend to be significantly longer than average in the *Anopheles* genome (e.g., at the level of coding sequence, transcript length, gene span as well as the number of exons, Figs 6A and S6B). Furthermore, we found a significantly higher fraction of *Drosophila* orthologues in the *msl-2* KO DE group than expected (Fisher's test $P = 6.34 \times 10^{-18}$) indicating that the *Anopheles* MSL-regulated genes tend to be conserved among Dipterans (Fig 6B, left panel). There was no difference between the expected and DE gene groups at the sequence level (% identity of the orthologues) (Fig 6B, right panel). We next analyzed the features of the proteins encoded by *msl-2* KO DE genes using the STRING database of known and predicted protein–protein interactions (Fig 6C) (Szklarczyk et al, 2018). We find that the proteins encoded by the Top400 DE genes (*msl-2* KO versus control) display significantly more protein–protein interactions than expected, indicating that they are highly interactive. If the targets regulated by MSL2 tend to frequently engage in protein–protein interactions and are part of multisubunit complexes, this may explain why their expression levels need to be tightly controlled by chromatin-regulatory pathways such as the H4K16ac–MSL pathway: poorly controlled expression could lead to stoichiometry imbalances, proteotoxicity and aggregation (Brennan et al, 2019).

Analysis of the functional domains of MSL2 targets with the Pfam database revealed an enrichment of "homeobox domain," which is a well-known module of developmental transcription factors, but also various other domains mediating processes occurring at the cellular surface (Fig 6D). We also performed Gene Ontology term

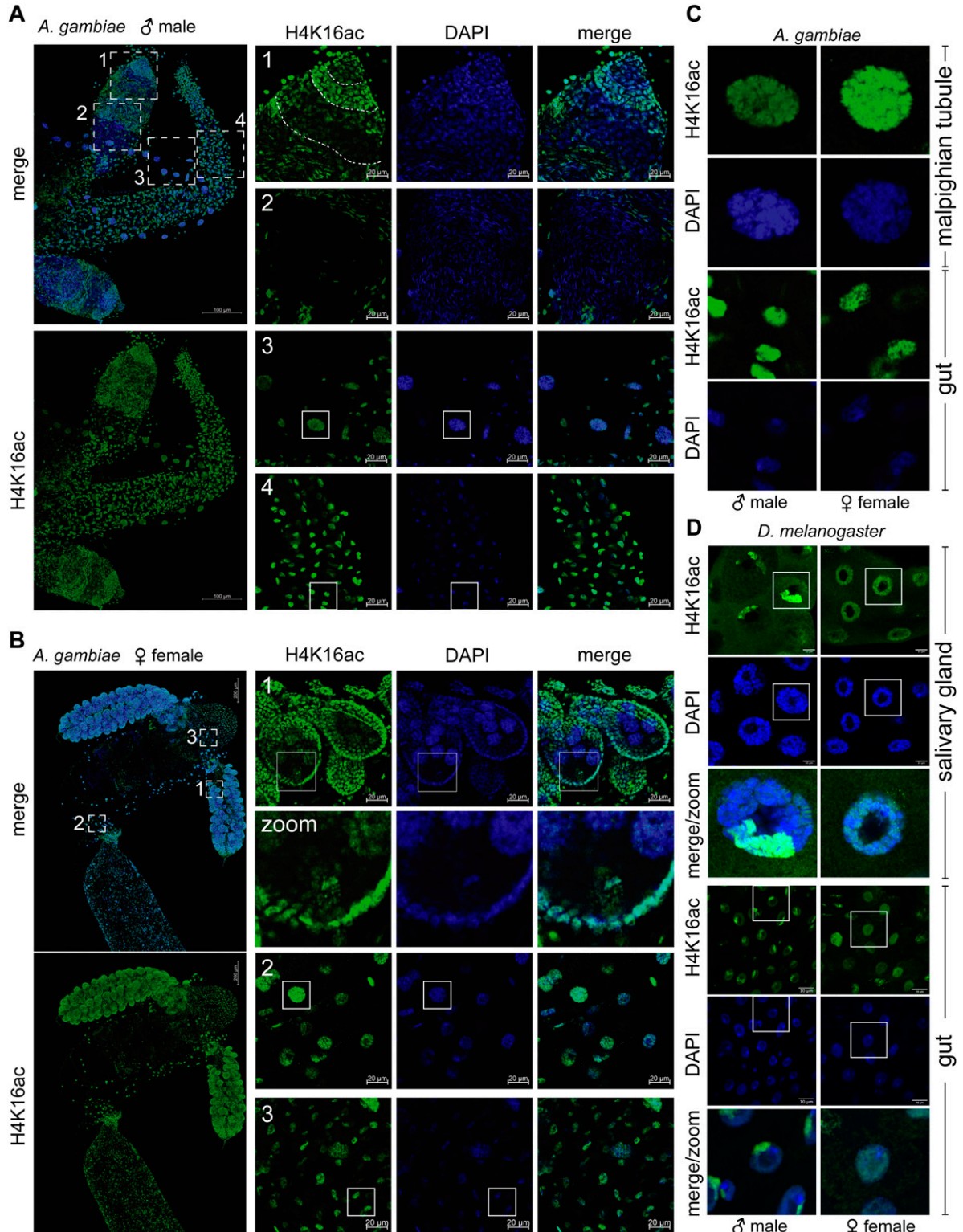

**Figure 4.  Tissue-specific distribution of Histone H4 Lysine 16 acetylation in *Anopheles gambiae*.**
**(A)** Immunofluorescence for H4K16ac (green) in adult male *A. gambiae* organs. The areas 1 and 2 highlight the testis, 3 the Malpighian tubules, and 4 the gut. The pictures are orthogonal projections of a z-stack with scale bar = 100 μm. The panels 1–4 on the right are single z-planes, where the highlighted areas were imaged at a higher magnification, scale bar = 20 μm. DAPI is shown in blue. **(B)** as in (A), but in adult female *A. gambiae* organs. The areas 1 highlight the ovary, 2 the Malpighian tubules and 3 the gut. The image labelled "zoom" shows a closeup of the ovarian area highlighted with the solid white rectangle (same image cropped and zoomed). **(C)** close-up of nuclei in Malpighian tubules (polyploid) or guts (diploid) that are highlighted with a solid white rectangle in (A, B). **(D)** Immunofluorescence for H4K16ac (green) in male and female *Drosophila* salivary glands (polyploid) and guts (diploid). The pictures are single z-planes with scale bar = 20 μm and DAPI shown in blue. The bottom panel shows a close-up of one nucleus in each males and females (same image zoomed).

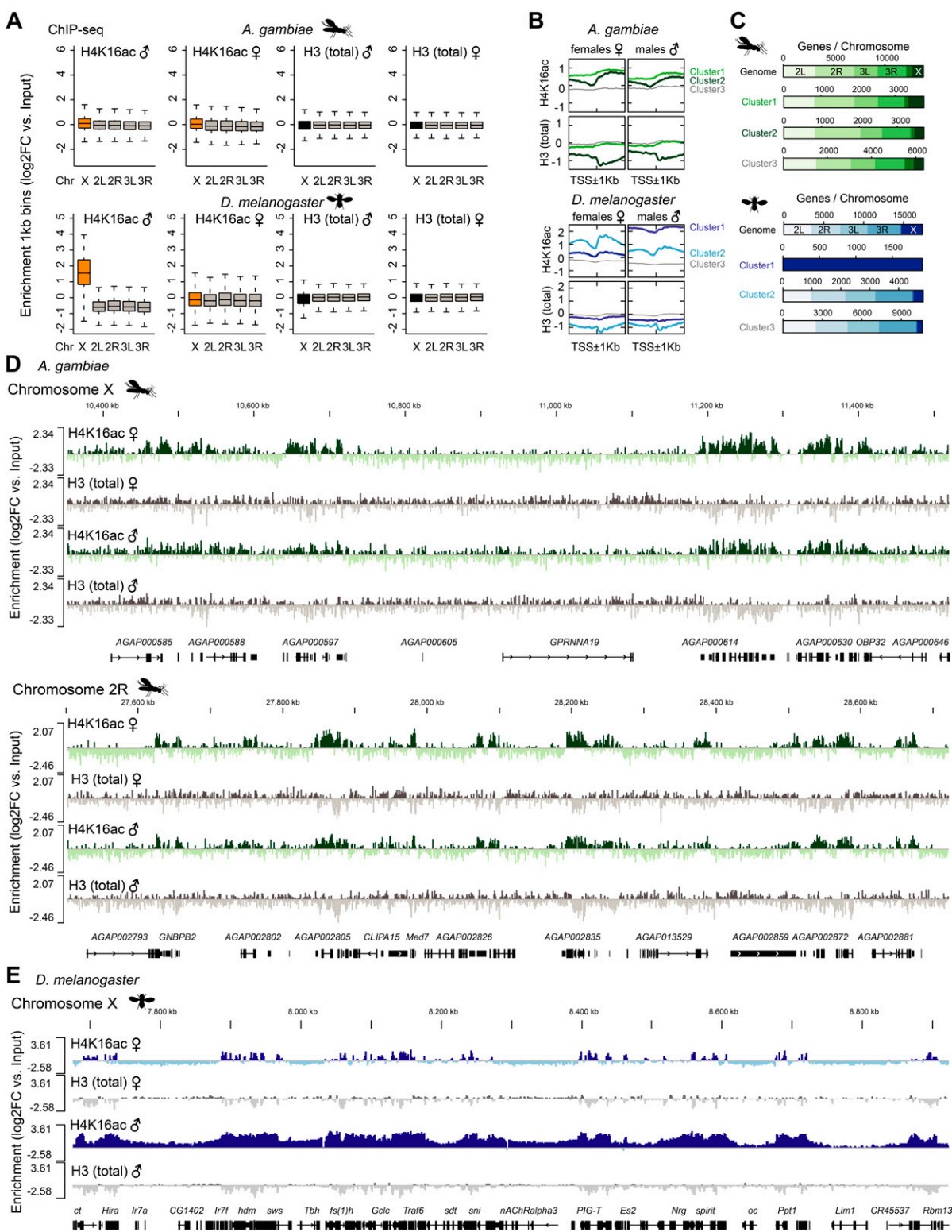

**Figure 5. Genome-wide profiles reveal H4K16ac on active genes, but no male-specific X chromosome enrichment.**
**(A)** ChIP-seq profiles for total Histone H3 and H4K16ac in *Anopheles gambiae* were generated from dissected adult midguts, in *Drosophila melanogaster* from L3 third instar larvae. Box plots show the mean $\log_2$(fold change) ChIP versus Input enrichments per 1 kb bin on each of the chromosomal arms. The biological replicates were merged (also see the Materials and Methods section). **(B)** as in (A), three unsupervised K-means clusters were generated from the $\log_2$(fold change) H4K16ac ChIP versus Input data. The transcription start site of each Agam or Dmel gene served as a reference point, while plotting the mean enrichment profile ± 1 kb. **(C)** as in (B), the bar plots show the chromosomal location of the genes present in clusters 1–3 compared to the overall number of genes on each of the chromosomal arms. **(D)** as in (A), genome browser snapshots of the normalized ChIP enrichment in Agam males and females on example regions of the X chromosome (top) and autosome (bottom). **(E)** as in (A), genome browser snapshots of the normalized ChIP enrichment in Dmel males and females on an example region on the X chromosome.

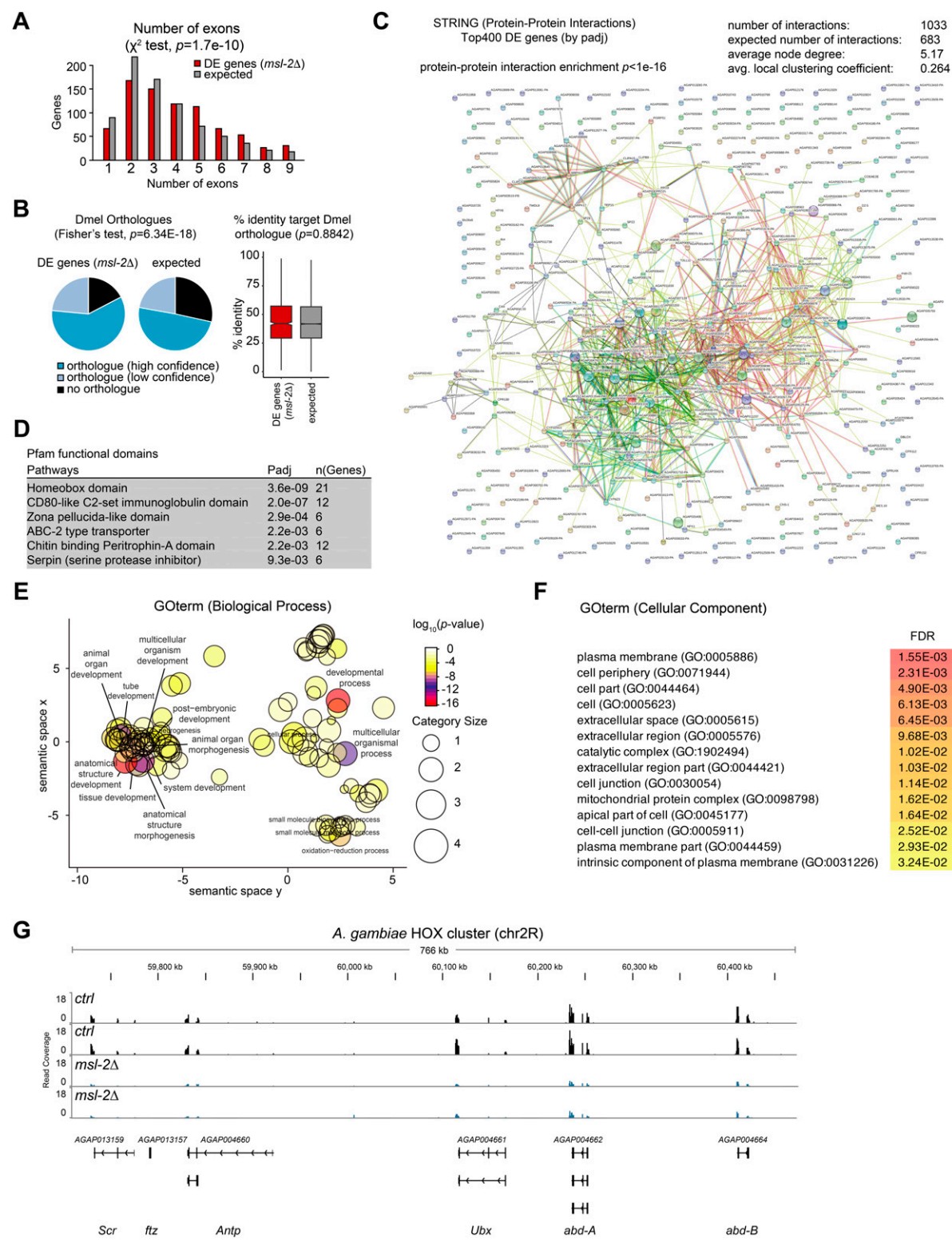

**Figure 6. Transcriptome analyses reveal a role for *Anopheles* MSL2 in regulating conserved developmental-regulatory genes.**

**(A)** RNA-seq as in Fig 3D and *n* = 926 *msl-2* DE genes in *A. gambiae* embryos (*msl-2* KO versus control, false-discovery rate < 0.05, 15 h after egg laying) were analyzed for overrepresentation concerning the numbers of exons versus the expected distribution using the ShinyGo (v0.61) gene-set enrichment tool (Ge et al, 2020). **(B)** RNA-seq as in Fig 3D, left: *msl-2* DE genes or all *A. gambiae* genes (expected) were analyzed for the presence of an Ensembl BioMart orthologue (black: no orthologue, light blue: low confidence orthologue, blue: high confidence orthologue). The statistical significance was evaluated by comparing underrepresentation in the "no orthologue" group (DE, *n* = 204 genes without orthologue) versus expected using a Fisher's exact test. Right: *msl-2* DE genes or all *A. gambiae* genes (expected) were analyzed for % of

analyses (Fig 6E and F), where we found misregulated biological processes such as "multicellular organism development," "tissue development," "tissue morphogenesis," or "cell differentiation" and cellular components such as "plasma membrane," "cell periphery," "extracellular space," or "cell junction." A prominent example in the down-regulated genes group was the *A. gambiae* HOX cluster (Fig 6G). HOX genes encode homeodomain-containing transcription factors that provide specificity and positioning information along the anterior-posterior axis and are conserved master regulators for setting up animal body plans (Gehring et al, 2009). Other mis-regulated genes include SPZ6 (chromosome 2L), a Spätzle-like cytokine involved in signalling and embryo patterning, or the OSIRIS family members Osi7, Osi9, and Osi14 (chromosome 2R), a conserved dosage-sensitive gene family encoding putative trans-membrane receptors. Collectively, this indicates that the MSL complex fine-tunes the expression of conserved genes that par-ticipate in highly intertwined developmental-regulatory networks orchestrating *Anopheles* embryogenesis. A perturbation of this function could provide a likely explanation for the early arrest of *msl-2* KO embryos (Fig 2).

## Discussion

In this study, we have explored the MSL complex–H4K16ac pathway and its involvement in X chromosome DC in the malaria mosquito *A. gambiae*. We find similarities, but also notable differences to the related Dipteran *D. melanogaster*, where these factors have been most extensively characterized. Zygotic *msl-2* KO *Anopheles* em-bryos fail to develop beyond the segmentation stage. In RNA-seq, we detect misexpression of key developmental-regulatory genes, including the Hox cluster. Genes affected upon *msl-2* KO tend to contain more exons, display a higher degree of conservation and participate in more protein–protein interactions than expected. Minor changes in the intertwined gene-regulatory networks op-erating in a highly precise manner during development can cu-mulatively lead to severe phenotypes, such as the ones observed by us and therefore explain why they require the fine-tuning by the MSL complex. Indeed, the fine-tuning of developmental-regulatory genes appears to be a common theme for MSL function in insects up to mammals (Valsecchi et al, 2018).

However, the time point of phenotype manifestation at only few hours of embryonic development in *Anopheles* is very distinct from the effect of MSL loss in *Drosophila* and mammals. In fruit flies, loss of *msl*s results in lethality occurring at the third instar larval stage (after around 5 d of development) (Belote & Lucchesi, 1980). The early *Anopheles* phenotype is also very different from the conse-quences of MSL3 loss in humans, which is associated with global developmental delay post birth affecting both male and female individuals (Basilicata et al, 2018). The reasons for these various time points of phenotype manifestation will need to be carefully dissected in future studies. They could, for example, be related to differences in maternally deposited factors that might allow faithful embryogenesis despite a zygotic loss, the presence of species-specific isoforms and paralogues or differences in stage-specific gene-regulatory networks in a given species.

In *Anopheles*, MSL2 is expressed in both males and females, and zygotic or germ line KO affects both sexes. Germ line–specific loss of *msl-2* results in sterility that is accompanied by severe atrophy of germline tissue. This is different from *Drosophila*, where MSL2 is male-specifically expressed, zygotic loss of *msl-2* is known to only affect males and KO females are fully fertile (Beckmann et al, 2005). If the fine-tuning of developmental genes appears to be the "an-cient" function of the MSL complex, how can it be achieved in *Drosophila* females, if the component that provides the chromatin targeting function to the complex (MSL2) is not present? We note that other MSL proteins, for example, MOF, can bind nucleic acids (Akhtar et al, 2000), making it possible that a subcomplex devoid of MSL2 (Samata et al, 2020) could be taking on these roles. In addition, other protein or ncRNA components could substitute for MSL2. One potential candidate is the protein CLAMP (Soruco et al, 2013), which aids recruitment of the MSL complex and is expressed in both sexes. Last but not least, some studies conversely suggest that MSL2 could indeed be expressed to some degree in females (Cheetham & Brand, 2018). Future studies will have to dissect the stage- and tissue-specific compositions and functions of the MSL complex in *Drosophila*.

The second intriguing question that arises is by which mecha-nism the MSL complex became co-opted to specifically target the male X chromosome in drosophilids. MSL2 is a DNA binding protein (Fauth et al, 2010) but the DNA binding CXC domain of MSL2 is dispensable for DC in vivo (Tikhonova et al, 2019). However, MSL2 also binds roX RNAs (Ilik et al, 2013), which are not present in species other than *Drosophila* (Quinn et al, 2016). Indeed, by virtue of a condensate-forming mechanism, MSL2-roX interaction governs stable association with the X (Valsecchi et al, 2021). The evolution of this species-specific RNA–protein interaction module therefore provides targeting to the X chromosome in *Drosophila*, but probably not other dipterans.

Another noteworthy aspect of our study is that X chromosome up-regulation in mosquitos is unlikely to be provided by the MSL-H4K16ac pathway. In our RNA-seq, loss of *msl*s in *Drosophila* is associated with global X chromosome down-regulation. However, an alternative model of inverse DC proposes that loss of *msl* genes induces autosomal up-regulation, instead of X down-regulation (Birchler, 2016). Indeed, we do observe a moderate up-regulation in our *Drosophila* data, which would be consistent with this idea. Whereas our datasets do not allow us to unambiguously distinguish

---

sequence identity in the target Dmel orthologue. The statistical significance was evaluated by comparing the % sequence identity for all orthologues (n = 971) versus expected using a Wilcoxon test. **(C)** RNA-seq as in Fig 3D, Protein–protein interaction network and statistical significance obtained on STRING (v11.0) (Szklarczyk et al, 2018) using the Top400 (by *Padj*) DE genes. **(D)** RNA-seq as in Fig 3D, enriched functional protein domains and statistical significance obtained from Pfam using the ShinyGo (v0.61) gene-set enrichment tool (Ge et al, 2020). **(E)** RNA-seq as in Fig 3D, Gene Ontology (GO) analysis of biological processes enrichment of *msl-2* DE genes. Enriched GO Terms were visualized with REVIGO. **(F)** RNA-seq as in Fig 3D, GO analysis of cellular components enrichment of DE genes upon *msl-2* KO in *A. gambiae*. **(G)** RNA-seq as in Fig 3D, Genome browser snapshots of the normalized read coverage in control (top, black) and *msl-2* KO embryos (bottom, blue) at the *A. gambie* HOX cluster on chromosome 2R. The two tracks represent two different biological replicates.

between these two models for *Drosophila*, our *Anopheles* dataset does not provide any evidence for skewed expression of an entire chromosome, neither the X nor autosomes. In addition, our ChIP-seq profiles show neither evidence of global X chromosome enrichment nor a sexually dimorphic pattern on a gene-by-gene level. Furthermore, H4K16ac does not form the typical X chromosome "territory" in immunofluorescence indicating that the deposition of this mark is not a universal feature for DC in Dipterans. This is reminiscent of another dipteran, *Sciara*, where *mle* and H4K16ac appear to play no role in DC (Ruiz et al, 2000). However, different from *Anopheles*, whose X chromosome is directly comparable to *Drosophila*, *Sciara* possesses a peculiar sex chromosome system undergoing imprinted elimination. The anole lizard, a reptile, displays enrichment of H4K16ac on the X chromosome of males compared to females (Marin et al, 2017). A study in monarch butterflies revealed a dichotomy of epigenetic marks on the neoZ chromosome, where both up-regulation (by H4K16ac) and dampening (by H4K20me1) DC mechanisms seem to operate concurrently in the same species (Gu et al, 2019). Yet in both cases, it remains unclear whether H4K16ac is deposited by the MSL complex or another enzyme. Multiple mechanisms have been proposed to be modulating gene-dosage imbalance ranging from transcription, RNA stability as well as protein turnover (Disteche, 2016). In light of the multitude of mechanisms involved in X chromosome regulation in different species, it will be interesting to further explore the epigenome of *A. gambiae* (Ruiz et al, 2021). Nevertheless, our data suggests that gene content appears not to be the main driver that governs the molecular pathway by which X chromosome DC is achieved. Elegant studies have demonstrated that Dipteran sex chromosomes show a high evolutionary turnover (Vicoso & Bachtrog, 2015). Our data adds on these findings and highlights that the complexity of dosage balancing mechanisms is probably much higher than previously anticipated. This fascinating diversity opens exciting opportunities for future evolutionary and mechanistic studies in Dipteran and other non-model organisms.

# Materials and Methods

### Mosquito rearing and transgenesis

*A. gambiae* mosquitos were maintained in standard insectary conditions (26–28°C, 75–80% humidity, and 12-/12-h light/dark cycle). Loss of function of *msl-2* (*AGAP000553*) was achieved by crossing transgenic mosquitos expressing Cas9 to transgenic partners expressing four different guide RNAs targeting exons of *msl-2*. The 20-nucleotide *msl-2*–specific sequences in the gRNAs were GATGAGTCCGCTGGGTGGTG, GAGCGTGAGCGTGGAGGAGG, GCGCAAGAGCCTCAGCTGTG, GTGGTAATGTTCACGATGCG. Each gRNA was cloned under the control of the *AGAP013557* U6 promoter in a cloning vector compatible with Golden Gate Cloning. The four gRNA modules were then assembled by Golden Gate Cloning in transgenesis plasmid pDSARN (Volohonsky et al, 2015). The full sequence of the resulting plasmid, pDSARN-4xMSL2gRNA can be provided upon request. Transgenic mosquitos were generated by inserting this plasmid in the Agam attP site of docking line X1 (Volohonsky et al, 2015). The resulting transgenic line was

called gMSL2 and made homozygous by COPAS sorting of neonate larvae carrying two copies of the DsRedNLS fluorescence transgenesis marker. As crossing partners of gMSL2 mosquitos, two different Cas9 parental lines were tested, both expressing Cas9 inserted in the X1 locus under the control of the germ line vasa promoter (Papathanos et al, 2009): one (YC9) encodes native Cas9 from *Streptococcus pyogenes* and carries a YFP transgenesis fluorescent marker; the other (eSpC9) is a Cas9 mutant less prone to off-target cutting (Slaymaker et al, 2016) and carries a DsRedNLS fluorescent marker. We initially attempted to generate heritable *msl-2* mutations by crossing Cas9-expressing to gRNA-expressing mosquitos to recover mutant progeny as described in Dong et al (2018). However, the lethality and sterility of F1 crosses prevented this approach. Instead, we produced *msl-2* knockout embryos by crossing Cas9 females with gRNA males. In this case, maternal deposition of active Cas9 in the oocyte and zygotic expression of gRNAs from the paternal copy of the U6-gRNA transgene resulted in 99% and 95% lethality using YC9 and eSpC9, respectively. The 1–5% escaper larvae presumably resulted from residual MSL2 function in embryos with incomplete mutagenesis or mutations preserving the gene's function. Most of these larvae died during development. Crossing Cas9 males to gRNA females produced normal F1 progeny, but CRISPR-Cas9 activity in the developing germ line resulted in atrophy of the male and female gonad and sterility of these F1 mosquitos. Note that when using *vasa-Cas9* fathers for crosses, we usually observe no significant contribution of paternal Cas9 protein via the sperm of the father.

### Polysome profiling

Flash-frozen male and female mosquitos were homogenized in 6 mM $Na_2HPO_4$, 4 mM $NaH_2PO_4$, 1% NP-40, 150 mM KCl, 5 mM $MgCl_2$ 1 mM EGTA, 50 mM NaF, 0.1 mM $Na_3VO_4$, 1 mM PMSF, 1× complete protease inhibitors, and 0.1 mg/ml cycloheximide and the extract was clarified by centrifugation. 140 OD (260 nm) of the extract were layered on a 15–60% sucrose gradient followed by centrifugation at 39,000 rpm for 3 h in a SW40 Rotor. After collecting the fractions, RNA was isolated from the fractions using phenol-chloroform extraction followed by isopropanol precipitation. RNA recovery was determined with a Qubit RNA assay. cDNA was synthesized from 100 ng RNA using the GoScript Reverse Transcription System (Promega) with random primers and subsequently quantified by qRT-PCR. The data were analyzed as described in Keller et al (2012) calculating the RNA enrichment relative to the total amount of RNA in a given fraction.

### Protein extracts and Western blots

Extracts from adult *Drosophila* were prepared by collecting 10 adult male or female flies aged 12–24 h after eclosion, which were decapitated, and heads were homogenized in 50 μl 1× Roti-Load. *Anopheles* extracts were prepared by bead-beating entire mosquitos in 1× Roti-Load. Samples were incubated for 5 min at 70°C and 10–20 μl used for separation by SDS–PAGE. Proteins were transferred overnight at 60 mA or 1 h 11 min at 111 V to polyvinylidene fluoride (PVDF) membranes using a Bio-Rad Wet Tank Blotting System in Tris-Glycine Transfer-Buffer with 10% methanol. The membranes were blocked for 30 min in 5% milk in PBS-0.1% Tween before incubation with primary antibodies in 0.5% milk PBS-0.1% Tween. Primary antibodies were used at the following dilutions: anti-HA (Mouse, #MMS-101P, 1:5,000; Covance), anti-MOF (Rabbit,

1:2,000 [Raja et al, 2010]), anti-MLE (Rat, 1:2,000 [Ilik et al, 2013]), anti-RNA Pol2 (Mouse clone 4H8, 101,307, 1:10,000; Active Motif), anti-MSL3 (Rat, 1:1,000 [Raja et al, 2010]), anti-MSL1 (Rabbit, 1:3,000 [Raja et al, 2010]), anti-Tubulin (Mouse, 44928, 1:5,000; Abcam), anti-RPB3 (Rabbit, 1:2,000 [Valsecchi et al, 2018]), anti-H4K16ac (Rabbit, 07-329, 1:3,000; Merck Milipore), anti–acetyl-Histone H3 (Rabbit, 06-599; Merck Milipore), anti-H3 (Mouse, 39763 1:5,000; Active Motif), and anti-H4 (61521, 1:1,000; Active Motif). Secondary antibodies (1:10,000) used were anti-mouse IgG HRP (NXA931), anti-rat IgG HRP (NA935V), and anti-rabbit IgG HRP (NA934) from Sigma-Aldrich. Blots were developed using Lumi-Light Western Blotting substrate (12015200001; Roche; Sigma-Aldrich) and imaged on a ChemiDoc XRS+ (Bio-Rad).

### RNA expression analysis

Samples were homogenized in TRIzol Reagent (15596026; Thermo Fisher Scientific) followed by RNA purification using a Direct-zol RNA MiniPrep kit or Direct-zol RNA Microprep (R2050 or R2062; Zymo Research) according to the manufacturer's instruction. cDNA was synthesized with the GoScript Reverse Transcription System (Promega) with random or oligo dT primers as indicated according to the manufacturer's instructions. For stranded polyA+ mRNA-Seq Library Preparation, the TruSeq stranded mRNA sample preparation kit (RS-122-2101; Illumina) was used.

### Quantitative real-time PCR

qPCR was performed on a Roche LightCycler II using FastStart Universal SYBR Green Master (04913914001; Roche) in a 7-$\mu$l reaction at 300 nM final primer concentration. Cycling conditions as recommended by the manufacturer were applied. We corrected for primer efficiency using serial dilutions.

### Semiquantitative RT–PCR

Semiquantitative RT–PCR was conducted with a one-step RT–PCR kit (210210; QIAGEN), where the reverse transcription with a gene-specific primer and subsequent PCR is conducted in a single reaction. 2.5 ng of RNA template was used in a total reaction mix of 10 $\mu$l as of the manufacturers' instructions. The cycling was conducted as follows: 30 min at 50°C, 15 min at 95°C (1 cycle); 40 s at 94°C, 40 s at 55°C, and 60 s at 72°C (28 cycles); 10 min at 72°C (1 cycle). Products were separated by 1.6% agarose gel electrophoresis in Tris-Borate buffer and stained with SYBR Safe–DNA Gel Stain (S33102; Thermo Fisher Scientific). Gels were scanned using a Typhoon FLA scanner (GE Healthcare Life Sciences).

### *Drosophila* rearing conditions

*D. melanogaster* were reared on a corn flour–molasses fruit fly medium (1 liter water, 12 g agar–agar threads, 18 g bakery yeast, 10 g soya flour, 80 g corn flour, 22 g molasses, 80 g malt extract, 2.4 g 4-hydroxibenzoic acid methylester [Nipagin], and 6.25 ml propionic acid) at 25°C, 70% relative humidity, and 12-h dark/12-h light cycle. All experiments were conducted at 25°C unless otherwise indicated. White-eyed ($w^{1118}$) Oregon-R or endogenously tagged *msl-2::HA* (Valsecchi et al, 2018) was used as a wild-type strain.

### Immunofluorescence and TUNEL staining

Tissues were stained according to standard procedures. In brief, tissues were fixed in 4% formaldehyde in PEM (0.1 M PIPES (pH 6.9), 1 mM EGTA, and 1 mM MgCl$_2$) for 20 min and washed three times with PBS. Samples were blocked for 1 h rocking with freshly prepared 0.5% BSA, 0.2% Triton X-100 in 1× PBS solution followed by overnight incubation with primary antibody (anti-H4K16ac [Rabbit, 07-329, 1:200; Merck Millipore], anti-H3K27me3 [Mouse, C15200181-50, 1:300; Diagenode]; staining validated also with anti-H3K27me3 [Rabbit, 9733, 1:250; Cell Signalling]). After Alexa fluorophore–labelled secondary antibody incubation (1:300), samples were thoroughly washed with PT (0.2% Triton X-100 in 1× PBS) before dissection of organs and mounting. Slides were mounted using ProLong Gold Antifade Mountant with DAPI (P36935; Thermo Fisher Scientific) and imaged using a Zeiss LSM880 Airyscan Microscope. For the TUNEL staining, embryos were dechorionated before fixation in 4% PFA. Embryos were washed three times for 5 min in PBS + 0.5% Triton X-100, followed by another three washes in PBS + 0.2% Tween-20. This was followed by TUNEL staining with the In Situ Cell Death Detection Kit, TMR red (12156792910; Sigma-Aldrich) following the procedures described in the manual. The embryos were mounted in ProLong Diamond Antifade Mountant (P36961; Thermo Fisher Scientific) and images obtained using a fluorescence spinning disc confocal microscope, VisiScope 5 Elements (Visitron Systems GmbH), which is based on a Ti-2E (Nikon) stand and equipped with a spinning disc unit (CSU-W1, 50 $\mu$m pinhole; Yokogawa). The set-up was controlled by the VisiView 5.0 software and images were acquired with a 20× air objective (CSI Plan Apo Lambda, NA 0.75; Nikon) and a sCMOS camera (BSI; Photometrics). 3D stacks of images were recorded for each sample, the exposure time was set to 200 ms for each channel. Maximum intensity projections of z-stacks and image processing were performed in Fiji (version 2.0.0-rc-68/1.52 h). The immunofluorescence stainings have been replicated in $n$ independent experiments: H4K16ac ($n = 4$ for both males and females), H3K27me3 ($n = 4$ males, $n = 2$ females for Cell Signalling Antibody; $n = 2$ for both males and females for mouse Diagenode antibody), TUNEL ($n = 2$).

## ChIP

ChIP was performed as described in Valsecchi et al (2018). In brief, midguts of *A. gambiae* were dissected in PBS and fixed in 0.2% formaldehyde for 15 min at RT. The fixation was stopped by addition of Glycine and the fixative was removed. Midguts were homogenized in buffer G1 (5 mM Hepes [pH 7.4], 20 mM KCl, 10 mM MgCl$_2$, 0.5% Tween-20, and 20% Glycerol containing 1 × Complete Protease Inhibitors) at 4°C. The homogenate was cleared through a sucrose cushion (G1 + 30% sucrose). Nuclei were washed once in (10 mM Tris [pH 8.0], 5 mM MgCl$_2$, 1 mM CaCl$_2$, 10 mM NaCl, 0.1% IGEPAL CA-630, and 0.25 M Sucrose) and then resuspended in Aline buffer (20 mM Tris [pH 8.0], 5 mM MgCl$_2$, 1 mM CaCl$_2$, 10 mM NaCl, 1% Triton X-100, and 0.25 M Sucrose). This was followed by MNase digestion at 25°C for 15 min typically using 0.5–1 $\mu$l Micrococcal Nuclease (M0247; New England Biolabs, amounts optimized depending on the batch). 10× High Salt Buffer (200 mM EDTA and 4 M NaCl) was added to stop the

**RT-qPCR primers**

| Anopheles gambiae gene | ID_fwd | Sequence | ID_rev | Sequence |
|---|---|---|---|---|
| msl-2 (exon1) | ck1235 | ACTTCAAGTGCAATTCCGGC | ck1236 | CACGGCAGTATCGAGTAGGC |
| msl-2 (exon2) | ck1239 | CCCACCATCAAAACCGTGTC | ck1240 | AAACTTGTTGCCACTTTCCGC |
| msl-2 (exon3) | ck1241 | CTACCTGCCCACAAACGTCA | ck1242 | TGCAAATACTGCGACCCGTA |
| mle | ck1642 | GCAAACCGATCATGGAAGCG | ck1643 | CCTGGCCCGAGTTGATGTAA |
| mof | ck1638 | AACAACCCAGCGAAGCAGATA | ck1639 | AAAAATCGTTCACCGCTCCCA |
| AGAP000634 | mb54 | TAAGCGCGGGAAGCTATAAA | mb55 | TTGTTGAAGCAGGAGCACAG |
| GPRMTH2 | mb46 | ACGGTGGAGTTCGTTACCTG | mb47 | CTGCTTCACCTTTTGCTTCC |
| Rps4 | mb36 | GCTGCCGCTGGTGATCTT | mb37 | TCGTCACCTCGCTGTTGGT |
| RpL32/Rp49 | ck1277 | GCTATGATAAACTCGCTCCCAA | ck1278 | TCATCAGCACCTCCAGCTC |
| RPL19 | ck1719 | AGCGTACCAAGGTTCGTGAG | ck1720 | GCGGTCTCCTCTTCCTTAGC |
| U6 snRNA | ck1253 | AGCATGGCCCTTTAAGTCAC | ck1254 | AGGCGTTGCTATGCCTTGAT |
| Hunchback | ck1275 | ACTACGCCACCAAGTACTGC | ck1276 | GGTGTACCGTCCAGGTTCAG |
| elovl | ck1279 | GCTTCATTGGCGTCAAGTACTTT | ck1280 | TCGCGGCCAGCATGTAGT |
| eve | ck1273 | CGGCTCCTATGCTAACCCAG | ck1274 | TGCTGACATGCTGTTTGCAC |
| Yob | ck1672 | CCCCTACCCAAGGTACGGAT | ck1673 | ACCTGATGGAACTGATACACGG |
| Yob (RT–PCR) | mb42 | CGCGCACTTGTTTATACTGTTAC | mb43 | CGAAAGGGAAAGTTACGAGC |
| Dsx (RT–PCR) | Dsx1F | AAAGCACACCAGCGGATCG | Dsx1R | CACCGAGATGTTCTCGTCC |
| ctrl1 (RT–PCR) | mb38 | ATCGCTATGGTGTTCGGTTC | mb39 | GCTGCAAACTTCGGCTATTC |
| ctrl2 (RT–PCR) | S7for | GGCGATCATCATCTACGTGC | S7rev | GTAGCTGCTGCAAACTTCGG |
| Drosophila melanogaster gene | ID_fwd | Sequence | ID_rev | Sequence |
| roX2 | q146 | GCCATCGAAAGGGTAAATTG | q147 | CTTGCTTGATTTTGCTTCGG |
| RpL32 | q148 | ATCGGTTACGGATCGAACAA | q149 | GACAATCTCCTTGCGCTTCT |
| msl-2 (RA) | q158 | CAATCCCGTCAGAGGTTGTAG | q159 | TCCACTGCTTCCTCCATTTC |
| msl-2 (RB) | q156 | ACAGTCACCTACACAAACCG | q157 | GCGTCCTCAAAATCTTTCTGC |
| msl-1 | q160 | GACGACGAAGATGACGAGAATAG | q161 | CTTCCTCAGTTCTGGCGTTTA |
| mle | q164 | TTGGACCTTCCTGTCGTAAAC | q165 | CATCCTGTTGCGAGGAATCTAT |
| mof | q166 | CACCCACGACGACATTATCTAC | q167 | AGGTGGTCTTGAATGGTCTTG |
| msl-3 | q168 | CCGGAGATGGCCGATTAAAG | q169 | CTCCTGTGGCACATGGTTATAG |

MNase reaction before treatment in a Bioruptor Pico (10 cycles, 30 s ON/OFF). After clarification by centrifugation (10 min at 12,000*g*), the supernatant was used for immunoprecipitation. 1 µl of anti–acetyl-Histone H4 (Lys16) Antibody, 07-329; Merck Millipore or 0.5 µl of Histone H3 antibody (mAb), MABI 0301 (39763; Active Motif) were used for immunoprecipitations followed by washes, elution by reverse cross-linking, RNase A, and Proteinase K treatment according to standard procedures. DNA was purified by phenol–chloroform extraction followed by ethanol precipitation. Libraries were made with a NEBNext UltraTM II DNA Library Prep Kit for Illumina (E7645; New England Biolabs) according to the manufacturer's instructions.

### Processing of RNA-seq and ChIP-seq datasets

For quality control, we used FastQC version: v0.11.3. For RNA-seq, paired-end reads were mapped with RNA STAR (Galaxy version 2.5.2b-0) to AgamP4_ensembl or dm6_ensembl with default parameters. Mapping statistics are reported in Table S1. The raw read counts per gene were obtained with featureCounts (1.4.6.p5). This was followed by differential expression analysis with DESeq2, R version 3.0.2. DESeq2 normalizes the read counts with a linear model, where counts for genes are modeled using a negative binomial distribution with fitted mean and a gene-specific dispersion parameter (details in Love et al [2014]). Table S1 reports statistical analyses for the *Drosophila* RNA-seq dataset for both the conventional DESeq2 standard normalization as well as where the data are normalized to the rRNA reads (note that rRNA species are typically present in RNA-seq data despite the polyA enrichment in the kit for mRNA-Seq). For the latter analysis, we also used DESeq2 and specified the rRNA genes read counts to be used for size factor estimation. For ChIP-seq, paired-end reads were mapped to AgamP4_ensembl with bowtie2 (Galaxy version v2.2.6.2) using default parameters. To generate coverage box plots for H4K16ac or H3 ChIP on each of the chromosomal arms, we used deepTools bamCompare v2.4.2 to calculate the readCount normalized $\log_2$ ratio versus the Input per 1 kb bin. To generate readCount normalized coverage bigwig files, we calculated the $\log_2$ ratio versus the Input using a bin size of 5 bp with deepTools bamCompare

v2.4.2. Duplicate reads and reads with a mapping quality <10 were removed, the X chromosome was ignored for scaling. The bigwig files were then used to generate k-means clusters with deepTools plotHeatmap v2.4.2, calculation of the mean enrichment for each gene using multi-BigwigSummary v2.4.2 and analysis with the IGV genome browser. GOterm analyses were conducted with the PANTHER Overrepresentation Test and Fisher analyses with Bonferroni correction on http://geneontology.org/ and visualized using REVIGO (http://revigo.irb.hr/r). Circos plots visualizing the homology between *A. gambiae* and *D. melanogaster* chromosomes were obtained with SyMAP Version v4.2 (default parameters). Previously published data were *Drosophila* H4K16ac ChIP-seq from (modEncode Consortium et al, 2010; Straub et al, 2013; Valsecchi et al, 2018; Samata et al, 2020), *A. gambiae* RNA expression data from Papa et al (2017) and Goltsev et al (2009). *D. melanogaster* modEncode RNA-seq data were obtained from FlyBase.

### Statistics

All statistics were calculated with R Studio (v1.3.959). In the box plots the line that divides the box into two parts represents the median, box bottom, and top edges represent interquartile ranges (IQR, $0.25^{th}$ to $0.75^{th}$ quartile [Q1-Q3]), whiskers represent Q1 – 1.5*IQR (bottom), Q3 + 1.5*IQR (top). Bar plots represent the mean ± SEM with overlaid data points representing independent experiments. Results were considered significant at FDR (*P*-value) below 0.05. ns, not significant; NA, not analyzed; SEM, standard error of the mean. Unless otherwise indicated statistical tests were two-sided.

### Protein sequences and alignments

Protein sequences were retrieved from FlyBase (http://flybase.org) and VectorBase (https://www.vectorbase.org/). Protein alignments were created on Clustal Omega (https://www.ebi.ac.uk/Tools/msa/clustalo/). Results of BLAST searches (BLASTP, 2.2.29+) are reported in Table S1. Lists of 1:1 orthologues were obtained via the Biomart tool on VectorBase.

### Bioinformatic and web resources

Bowtie2 (https://github.com/BenLangmead/bowtie2) (Langmead & Salzberg, 2012); deepTools2 (https://deeptools.readthedocs.io/en/latest/) (Ramírez et al, 2016); Galaxy (https://github.com/bgruening/galaxytools) (Grüning et al, 2017); IGV (https://software.broadinstitute.org/software/igv/) (Robinson et al, 2011); R (https://www.r-project.org); DESeq2 (http://bioconductor.org/packages/DESeq2/); GoTerm analysis (http://geneontology.org/) (The Gene Ontology Consortium, 2019); REVIGO (http://revigo.irb.hr/r) (Supek et al, 2011); STRING (https://string-db.org/) (Szklarczyk et al, 2018); ShinyGO (http://bioinformatics.sdstate.edu/go/) (Ge et al, 2020).

## Data Availability

RNA-seq and ChIP-seq data have been deposited to the Gene Expression Omnibus under the accession number GSE153780. All other relevant data supporting the key findings of this study are available within the article and its supplementary files or from the corresponding authors upon request.

## Supplementary Information

## Acknowledgements

We thank the Max Planck Institute of Immunobiology and Epigenetics facilities and the IMB Microscopy Core Facility for support. We thank Marc Bayer and Sarah Toscano for contributing initial analyses to set up the project, as well as Devon Ryan and Gina Renschler for help with data processing and bioinformatic analyses. We thank Gina Renschler, Maria Shvedunova, and Agata Kalita for their input on the manuscript. CI Keller Valsecchi was supported by a Human Frontier Science Program long-term fellowship (000233/2014-L). This study was supported by the German Research Foundation (DFG) under Germany's Excellence Strategy (CIBSS–EXC-2189–Project ID 390939984). Mosquito production and insectarium operation were supported by Agence Nationale de la Recherche grant #ANR-11-EQPX-0022. This work was supported by the DFG under the CRC 992 (project A02) and CRC 1381 (project B3) awarded to A Akhtar.

### Author Contributions

CI Keller Valsecchi: conceptualization, data curation, formal analysis, supervision, funding acquisition, validation, investigation, visualization, methodology, project administration, and writing—original draft, review, and editing.
E Marois: conceptualization, resources, funding acquisition, validation, investigation, visualization, methodology, and writing—review and editing.
MF Basilicata: formal analysis, supervision, validation, investigation, visualization, methodology, project administration, and writing—review and editing.
P Georgiev: conceptualization, resources, investigation, methodology, project administration, and writing—review and editing.
A Akhtar: conceptualization, resources, supervision, funding acquisition, methodology, project administration, and writing—review and editing.

### Conflict of Interest Statement

The authors declare that they have no conflict of interest.

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
