## [Reviewer comments · Life Science Alliance]

Life Science Alliance

Distinct mechanisms mediate X chromosome dosage compensation in *Anopheles* and *Drosophila*

Claudia Keller Valsecchi, Eric Marois, M. Basilicata, Plamen Georgiev, and Asifa Akhtar

DOI: <https://doi.org/10.26508/lsa.20200996>

Corresponding author(s): Asifa Akhtar, Max Planck Institute of Immunobiology and Epigenetics and Claudia Keller Valsecchi, Institute of Molecular Biology (IMB), Mainz, Germany

Review Timeline:

Submission Date:	2020-12-16
Editorial Decision:	2021-02-07
Revision Received:	2021-06-03
Editorial Decision:	2021-06-18
Revision Received:	2021-06-27
Accepted:	2021-06-28

Transaction Report:

February 7, 2021

Re: Life Science Alliance manuscript #LSA-2020-00996-T

Asifa Akhtar
Max Planck Institute of Immunobiology and Epigenetics

Dear Dr. Akhtar,

Thank you for submitting your manuscript entitled "Distinct mechanisms mediate X chromosome dosage compensation in Anopheles and Drosophila" to Life Science Alliance. The manuscript was assessed by expert reviewers, whose comments are appended to this letter.

As you will note from the reviewers comments below, both reviewers are overall positive about the manuscript, but have raised some interesting and important points that need to be responded to prior to further consideration of the manuscript. We thus encourage you to submit a revised manuscript to us that addresses all of the reviewers' points, along with a point-by-point rebuttal to the reviewers' comments.

Thank you for this interesting contribution to Life Science Alliance. We are looking forward to receiving your revised manuscript.

Sincerely,

Shachi Bhatt, Ph.D.

Executive Editor

Life Science Alliance

<https://www.lsa-journal.org/>

Interested in an editorial career? EMBO Solutions is hiring a Scientific Editor to join the international

Life Science Alliance team. Find out more here -
https://www.embo.org/documents/jobs/Vacancy_Notice_Scientific_editor_LSA.pdf

- A letter addressing the reviewers' comments point by point.
- An editable version of the final text (.DOC or .DOCX) is needed for copyediting (no PDFs).
- High-resolution figure, supplementary figure and video files uploaded as individual files: See our detailed guidelines for preparing your production-ready images, <https://www.life-science-alliance.org/authors>
- Summary blurb (enter in submission system): A short text summarizing in a single sentence the study (max. 200 characters including spaces). This text is used in conjunction with the titles of papers, hence should be informative and complementary to the title and running title. It should describe the context and significance of the findings for a general readership; it should be written in the present tense and refer to the work in the third person. Author names should not be mentioned.

B. MANUSCRIPT ORGANIZATION AND FORMATTING:

Reviewer #1 (Comments to the Authors (Required)):

This study examines the possibility that the dosage compensation of the X chromosome in male mosquitoes is similar or dissimilar to that which occurs in *Drosophila*. A knockout of the *msl-2* gene in mosquito led to lethality in both sexes as opposed to only males in *Drosophila*. Transcriptomic analysis in knockout embryos showed no evidence of skewed expression expected with an upset in gene expression differentially on different chromosomes. The data do support the contention that *Drosophila* and mosquito are different.

Nevertheless, I do not subscribe to the interpretation afforded the data for *Drosophila*. I have previously communicated with this laboratory that it is unlikely that destruction of the MSL complex leads to an overall reduction of the X expression but does lead to an overall upregulation of the

autosomes (with exceptions of course). I reviewed the Conrad et al paper from this group that was published in Science when it was originally submitted to Nature. Two of the three reviewers noted that the data were consistent with an overall increased PolII occupancy on the autosomes with little reduction on the male X. My review of the revised version of that paper for Nature was sent to the editor for reference. Note that I recommended acceptance if the proper interpretation was presented but the editor chose to do otherwise. In the current submission, the data in Figure 3B are consistent with this view. The program used for DE detection automatically normalizes to transcriptome size and it is not possible to detect such a change, which might be the case if the autosomes were globally upregulated about two fold (See Hou et al 2018 PNAS 115: 113321-11330 for why this is the case). Figure 2B shows a skewing of normalized counts downward for X linked genes but there is also a skewing upward for autosomal genes. As far as can be determined from the manuscript, the analysis would basically normalize to the autosomes and so the skewing for them is less obvious but would be stronger for the X. There was no apparent attempt to control for global effects as a whole such as using a spike-in to total RNA before preparation for RNAseq, which could potentially distinguish these alternatives (if total rRNA changes also, then it cannot). The two fold increase of the autosomes would occur due to an inverse effect on expression of the X to A stoichiometry. Metafemales have 3X:2A and have X dosage compensation in the absence of MSL2 and down regulation of the autosomes (Sun et al 2013 PNAS 110: 7383-7388 and references within). When the flipside of removing the MSL complex is done in males, the X is dosage compensated and the autosomes would be increased. This group has ignored the empirical observation of the inverse effect over the years but a dosage series of the sex chromosomes in human has recently revealed a similar effect (Raznahan et al 2018 PNAS 115: 7398-7403 and Zhang et al 2020 PNAS 117: 4864-4873) probably due to a dosage series of the small pseudoautosomal region. In addition, several studies have failed to find an increase in expression when the MSL complex is targeted to the X chromosome in females or to autosomal transgenes in males (Sun et al 2013 PNAS 110: 808-817 and references within) providing no evidence that it causes upregulation but rather a constraining activity on H4Lys16. This reviewer recently saw a preprint verifying these results using proper normalization protocols and validating the results with in situ FISH.

In a more general sense, it is now clear that most transcription factors exhibit a dosage effect on their target genes. Indeed, the whole basis of co-expression networks relies on this principle. If there were indeed a down regulation of the X chromosome in msl minus males, it is inconceivable that there would not be global modulations of target genes across the genome.

Given the history of this group, I am sanguine about the possibility that they would open their minds to other possibilities than their own Baconian Idols of the Cave, yet one can only hope. Given that this report comes from a Max Planck Institute, one might reflect on a quote from Max Planck: "A new scientific truth does not triumph by convincing its opponents and making them see the light, but rather because its opponents eventually die." I find this sentiment rather dark and wish no one any ill will. Indeed, I would welcome a dialogue based on empirical data.

James A. Birchler

Reviewer #2 (Comments to the Authors (Required)):

The manuscript by Valsecchi et al., entitled "Distinct mechanisms mediate X chromosome dosage

compensation in *Anopheles* and *Drosophila*" studied X chromosome dosage compensation regulation in *Anopheles gambiae* mosquitoes. They found that Agam has a distinct dosage compensation mechanism, which does not depend on H4K16ac target male X and *msl2* does not play a sex-specific role as in *Drosophila* DC. Overall, these results are of interest in dosage compensation mechanism across species. However, there are several issues that need to be addressed before strong conclusions can be drawn.

Main points

Quality of RNA-seq data produced in early Agam embryos in wt and germline *msl-2* KO phenotypes. 1). Authors only used 1 or 2 replicates in their time points. It is not robust enough in the RNA-seq for DEG callings and downstream *msl2* functional analysis (in Fig.6) as well as some strong conclusions being made in the manuscript. 2). The authors did not report details in their material and method of how many million reads they sequenced per sample, the mapping rates, and the Pearson correlation coefficient value among replicates for data quality/reproducibility assessment.

It is unclear when does dosage compensation occur in Agam embryos. In *msl-2* KO embryos, it is difficult to tell when do they arrest in the development stages. Do male/female embryos have different development rate, and do they all passed ZGA and whether X dosage compensation is initiated? Have you done any co-IP to see if MSL complex has formed yet? Maybe some immunostainings of *msl* proteins in early embryos?

The observation of H4K16ac distributed equally across chromosomes reminds me of another dipteran *Sciara ocellaris* and its MSL complex equally distributed on X chromosome and autosomes (Ruiz et al 2000). This could be a conserved mechanism within Dipteran Nematocera suborder species? The author should consider discussing this in their discussions.

Specific comments:

Fig. 1B. Do these synteny blocks pairs generated over genomic sequences or just for ortholog genes?

Fig 1E and Fig. S1 B, C, D. The qPCR and mRNA-seq data presented in these 4 plots did not provide any statistical test to support their claim on sex-biased differential expression. For example, the authors state "we found a modestly biased expression of *msl2*, *msl3*, *mod* and *mle* in males", and "In RNA-seq dataset, Agam *msl*s we all classified as not sex-biased." To just visually access (without any p-value from test), these *msl* genes show higher expression in males in qPCR data but a higher expression in females in mRNA-seq data. Have you done any statistical test to support your statement of biased/not biased expression? What is the normalization used in RNA-seq expression value, are they normalized by read length and library depth? I also will not use the word "validated", first these data are generated from different tissues, and the results are not consistent. Moreover, there is no qPCR data on these *msl* genes in *Dmel* as a comparison, does it also show such male-biased in these *msl* genes?

Fig. 1F and Fig S1E, are these sexed embryos? If they are pooled sex-mixed embryos, the observation of *msl* genes mRNA drop along development does not mean much regarding to male X dosage compensations.

Does Agam have orthologous to roX RNAs?

Fig 2B and Fig S2A. When do they stop development in *msl2*-KO embryos? Do they pass Zygotic genome activation? The embryos seem dead at 30h, do you know if they still viable in 40h and 46h? Is there any way to assess this?

Fig. 2E. What stage are these *msl2*-KO embryos? Do you have some immunostainings data of *msl* proteins in early embryos?

Fig. 2F. Here is only for females, any data/image on F1 male testis atrophy?

Fig. 3 and Fig 3s. What stage are these *Drosophila* MSL2-knockdown embryos? Are they comparable to 6-7hr Agam embryos regarding the developmental stage? One of the reasons you don't have as many DEGs in Agam dataset could be that you have fewer replicates (1 or 2) and from mixed time points compared to in *Drosophila* you have four replicates. What is the Pearson correlation value among Agam embryos?

Fig 3D, with only 2 replicates in 15hr Agam embryos compare to 4 replicates in *drosophila*, I won't be too surprised that you get a smaller number of DEGs. I also don't understand this sentence "the majority of downregulated targets (523 of 566) were affected by at least 75% or more". Where is this 523 from? And what is 75% relative to?

Fig. 4 and Fig S4. How do you localize the histone marks to the X chromosome in these 2 figures? Have you done X chromosome DNA-FISH to see if you can have X chromosome territory overlap with these histone marks?

Fig. 4C-D, Do these 2 tissue comparable since the salivary glands have polytene chromosomes? Can you show H4K16ac in *Drosophila* Malpighian as a better comparison? Or staining H4K16ac in Agam - do they have polytene chromosomes? It seems conserved.

Fig. 5. Why compare ChIP-seq generated from Agam midgut to *Dmel* L3 Larvae? Shouldn't Larvae in both groups be a better comparison?

Fig. S6. Is Student's t-test the right one? A non-parametric test might be better here for distribution.

Fig. 6A, Did this statistical test between DE genes and expected as 2 groups, or it can be done pairwise in each exon numbers. If so you can put an asterisk on top of exon number 4 to 9.

Fig. 6G. Need to label the sample group on the left of the genomic track.

Life Science Alliance manuscript #LSA-2020-00996-T

We thank both reviewers for their constructive criticism, which have helped to improve our manuscript. In addition to the point-by-point response (below), we provide a brief overview of the new experiments and analyses added upon revision:

Fig. 1 and Fig S1	Statistical analyses added.
Fig. S1D	qPCR showing RNA levels of msl -genes in male and female Drosophila .
Fig. 2C and Fig. S2B	TUNEL staining describing the onset of embryo death in msl-2 KO embryos.
Fig. 3	Immunofluorescence of D. melanogaster added to complete panels. Color changed for consistency.
Fig. S2D	Picture of msl-2 KO F1 testis.
Fig. S3A	Euclidian distances for D. melanogaster RNA-seq.
Fig. S4C	Genome-browser snapshot of different D. melanogaster H4K16ac ChIP-seq datasets (Micrococcal nuclease and sonication fragmentation).
Table S1	Added: RNA-seq mapping statistics, statistical analyses for qPCR and RNA-seq

In addition, we have clarified and expanded the methods, where necessary and amended our text.

Reviewer #1

This study examines the possibility that the dosage compensation of the X chromosome in male mosquitoes is similar or dissimilar to that which occurs in *Drosophila*. A knockout of the *msl-2* gene in mosquito led to lethality in both sexes as opposed to only males in *Drosophila*. Transcriptomic analysis in knockout embryos showed no evidence of skewed expression expected with an upset in gene expression differentially on different chromosomes. The data do support the contention that *Drosophila* and mosquito are different.

Nevertheless, I do not subscribe to the interpretation afforded the data for *Drosophila*. I have previously communicated with this laboratory that it is unlikely that destruction of the MSL complex leads to an overall reduction of the X expression but does lead to an overall upregulation of the autosomes (with exceptions of course).

We thank the reviewer for his thorough summary and analysis of our data. We also appreciate that the reviewer thinks that our “data support the contention that *Drosophila* and mosquito are different”. The point of data interpretation (X downregulation vs. autosome upregulation) in *Drosophila* certainly deserves attention and we discuss this issue now in further detail below as well as in the manuscript.

I reviewed the Conrad et al paper from this group that was published in *Science* when it was originally submitted to *Nature*. Two of the three reviewers noted that the data were consistent with an overall increased PolIII occupancy on the autosomes with little reduction on the male X. My review of the revised version of that paper for *Nature* was sent to the editor for reference. Note that I recommended acceptance if the proper interpretation was presented but the editor chose to do otherwise.

We acknowledge the comment regarding the 2012 *Science* study, but would like to emphasize that this is not the point of this current 2021 manuscript focused on *Anopheles*. Even if interpretations diverge with regards to the *Drosophila* data (see below), this does not change the major take home message of our manuscript: Mosquitos have another mechanism and do not show a difference between X and autosomal gene expression in *msl-2* knock-outs.

In the current submission, the data in Figure 3B are consistent with this view. The program used for DE detection automatically normalizes to transcriptome size and it is not possible to detect such a change, which might be the case if the autosomes were globally upregulated about two fold (See Hou et al 2018 PNAS 115: 113321-113330 for why this is the case). Figure 2B shows a skewing of normalized counts downward for X linked genes but there is also a skewing upward for autosomal genes. As far as can be determined from the manuscript, the analysis would basically normalize to the autosomes and so the skewing for them is less obvious but would be stronger for the X. There was no apparent attempt to control for global

effects as a whole such as using a spike-in to total RNA before preparation for RNAseq, which could potentially distinguish these alternatives (if total rRNA changes also, then it cannot).

We would like to point out to the reviewer that this is a mixed-sex experiment, where a 50:50 population of male and female *Drosophila* embryos is analyzed (reason: to be equivalent to the *Anopheles* experiment, where sexes cannot be distinguished). Since MSL2 is not expressed in *Drosophila* females (Kelley et al. 1995), its deletion is not expected to cause any effects in females. Half of our RNA-seq reads (the ones coming from the females) in each sample (control, *msl-2* KO), are therefore unaffected / not skewed and are helpful for proper normalization of the data (they could be regarded as a “spike”). The analyses provided upon revision do indeed suggest that even the conventional normalization does not skew our data (see below).

To address the reviewer's comment, we have now applied an alternative way of normalization to address the concern of skewing by global autosome misregulation. For this, we have used transcripts that are not produced by RNA Polymerase 2 for normalization, as they are unaffected by changes in the MSL/DC pathway (rRNA, snoRNA, snRNA). rRNA has been previously used as a loading control for Northern blots in (Sun and Birchler 2009; Sun et al. 2013)) and are contained in our data, since the poly(A) enrichment during library preparation is never 100% complete leaving some reads against those RNA species for normalization.

The results and comparison of both normalizations are presented in the following Figure for the reviewer. As in the manuscript Fig. 3C/F, the plots show the density of the log₂FC of all genes, irrespective of whether they are scored as differentially expressed (DE) or not.

We find that:

1. No matter the normalization (Standard DESeq2 or rRNA), the *Drosophila* X chromosome is significantly downregulated in *msl-2* KO. The X is the most highly affected chromosome, both by p-value as well as by fold-change.
2. No matter the normalization, the *Drosophila* autosomes are moderately upregulated in *msl-2* KO. Fold change and significance are lower compared to the changes affecting the X.
3. The log₂FC distributions of autosomes compared to the X chromosome are significantly different (black lollipop line on the right for each pairwise comparison).
4. In *Anopheles*, both X and autosomal densities are not globally changed (they center at 0), indicating that some genes are up-, others are downregulated but there are no global, chromosome-wide effects.

5. In *Anopheles*, the log₂FC distributions on autosomes and on the X chromosome do not differ from each other (black lollipop line on the right of the tables for each comparison).

We have added these statistical analyses in Table S1 and also expanded the discussion on autosome upregulation. We hope that this answers the concern of the reviewer.

Drosophila

Standard DESeq2 normalization

Normalization	Standard			
	mean log ₂ FC	median log ₂ FC	test	p-value
Chromosome				
2L	0.04714944	0.01677698	>0	1.78E-06
2R	0.05929847	0.03066517	>0	3.03E-14
3L	0.07210015	0.02490367	>0	2.19E-15
3R	0.08340172	0.0318764	>0	1.53E-28
X	-0.1579672	-0.2136555	<0	7.50E-149

Non-Pol2 transcripts normalization

Normalization	non-Pol2			
	mean log ₂ FC	median log ₂ FC	test	p-value
Chromosome				
2L	0.05243852	0.0499239	>0	7.63E-10
2R	0.1196257	0.06723423	>0	1.29E-24
3L	0.1653776	0.06151202	>0	4.08E-30
3R	0.1451806	0.06846313	>0	7.63E-45
X	-0.1417605	-0.2489197	<0	1.45E-79

Anopheles

Standard DESeq2 normalization

Normalization	Standard			
	mean log ₂ FC	median log ₂ FC	test	p-value
Chromosome				
2L	-0.00974869	0.0136351	>0	0.088
2R	-0.04530152	-0.004589814	>0	0.952
3L	-0.0044336	0.001504903	>0	0.379
3R	-0.0027907	0.02416154	>0	0.010
X	-0.05904444	-0.01732854	<0	0.017

Reviewer Figure: (Top and middle) RNA-seq experiments were conducted on *D. melanogaster* F1 embryos obtained from crosses of *msl-2^Δ/CyO-GFP* fathers with *msl-2^Δ/CyO-GFP* mothers. Non-fluorescent *msl-2^Δ/msl-2^Δ* null mutant embryos (abbreviated as *msl-2*) were compared with fluorescent *msl-2^Δ/CyO-GFP* controls. Density plots of the log₂[fold change] obtained with DESeq2 for genes on each of the indicated chromosomal arms. All genes were taken into account for the analyses, irrespective of whether they are scored as DE or not. The tables on the right report the results of a one-sided Wilcoxon signed rank test with continuity correction, where the column “test” indicates whether the p-value was obtained for the alternative hypothesis greater (>0) or less (<0). The lollipops report the p-values from a two-sided Wilcoxon rank sum test with continuity correction obtained by comparisons between X and autosomal distributions. (Top) Default normalization according to DESeq2. (Middle) Normalization of the same data only using reads mapping to rRNA, snRNA, snoRNA. (bottom) Analysis and statistical tests obtained for the *Anopheles* RNA-seq dataset (Embryos 15h after egg laying, control versus *msl-2* KO).

The two fold increase of the autosomes would occur due to an inverse effect on expression of the X to A stoichiometry. Metafemales have 3X:2A and have X dosage compensation in the absence of MSL2 and down regulation of the autosomes (Sun et al 2013 PNAS 110: 7383-7388 and references within). When the flipside of removing the MSL complex is done in males, the X is dosage compensated and the autosomes would be increased. This group has ignored the empirical observation of the inverse effect over the years but a dosage series of the sex chromosomes in human has recently revealed a similar effect (Raznahan et al 2018 PNAS 115: 7398-7403 and Zhang et al 2020 PNAS 117: 4864-4873) probably due to a dosage series of the small pseudoautosomal region.

Our experiments are not designed to unambiguously answer this question, since this is not the point of our paper. Future studies that may contain e.g. the spiked-in normalization or UMI-based quantifications with appropriate controls and analyze wild-type as well as other genetic scenarios will need to be performed to ultimately address this issue. Such studies should also address the interesting effects seen in metafemales.

In addition, several studies have failed to find an increase in expression when the MSL complex is targeted to the X chromosome in females or to autosomal transgenes in males (Sun et al 2013 PNAS 110: 808-817 and references within) providing no evidence that it causes upregulation but rather a constraining activity on H4Lys16.

We would like to point out that we and others have seen upregulation of autosomal transgenes in males (e.g. (Alekseyenko et al. 2008; Ramírez et al. 2015; Park et al. 2002)) as well as X-linked genes upon ectopic expression of MSL2 in females (e.g. (Valsecchi et al. 2018), Figs. 4g, 1b/c, S1d) via deposition of H4K16ac. These are independent findings from different, well established labs in the field that cannot be ignored.

This reviewer recently saw a preprint verifying these results using proper normalization protocols and validating the results with in situ FISH.

Without a reference / DOI, we can unfortunately not relate to which preprint the reviewer is referring to.

In a more general sense, it is now clear that most transcription factors exhibit a dosage effect on their target genes. Indeed, the whole basis of co-expression networks relies on this principle. If there were indeed a down regulation of the X chromosome in *msl* minus males, it is inconceivable that there would not be global modulations of target genes across the genome.

We have never claimed that there are no modulations of other genes across the genome in *msl-2* KO males. There are 797 DE genes on X but also 465 (chr2L), 568 (chr2R), 362 (chr3L)

and 543 (chr3R), respectively, DE genes on autosomes. This had already been reported in Fig 3B (and also in (Valsecchi et al. 2021)) and is obvious by looking at the graph/plots.

Given the history of this group, I am sanguine about the possibility that they would open their minds to other possibilities than their own Baconian Idols of the Cave, yet one can only hope. Given that this report comes from a Max Planck Institute, one might reflect on a quote from Max Planck: "A new scientific truth does not triumph by convincing its opponents and making them see the light, but rather because its opponents eventually die." I find this sentiment rather dark and wish no one any ill will. Indeed, I would welcome a dialogue based on empirical data.

James A. Birchler

We thank the reviewer, Prof. James Birchler, for his stimulating and thoughtful comments and would like to indicate that we are open to a critical dialogue on the way dosage compensation in *Drosophila* is achieved. However, we are not convinced that the current manuscript centered on *Anopheles* is the right format for such a critical dialogue but would rather favor a different approach (that potentially could involve others in the field too).

We have now amended the discussion paragraph with regards to the autosome upregulation and also provided the statistics/validation of the RNA-seq analysis (see above) in Table S1. We hope that this answers the concerns of the reviewer and that our paper is now ready for publication.

Reviewer #2

The manuscript by Valsecchi et al., entitled "Distinct mechanisms mediate X chromosome dosage compensation in Anopheles and Drosophila" studied X chromosome dosage compensation regulation in *Anopheles gambiae* mosquitos. They found that Agam has a distinct dosage compensation mechanism, which does not depend on H4K16ac target male X and msl2 does not play a sex-specific role as in *Drosophila* DC. Overall, these results are of interest in dosage compensation mechanism across species.

We thank the reviewer for their positive and supportive evaluation and summary of our manuscript.

However, there are several issues that need to be addressed before strong conclusions can be drawn.

Main points

Quality of RNA-seq data produced in early Agam embryos in wt and germline msl-2 KO phenotypes. 1). Authors only used 1 or 2 replicates in their time points.

We apologize, if we have been unclear in our description: We have 3 replicates for the 6-7h timepoints and 2 replicates for the 15h timepoint. It is true that we have only produced a single replicate for the 24h timepoint, but we have not performed any DEG analysis on this time point but only provided the Euclidian distance compared to the other samples (Supp. Fig. S3A). We have now added more details in Table S1 and methods to clarify this issue, which also includes the quality metrics requested below.

It is not robust enough in the RNA-seq for DEG callings and downstream msl2 functional analysis (in Fig.6) as well as some strong conclusions being made in the manuscript.

We have used DESeq2 (Love et al. 2014) for DEG calling, which is compatible with the analysis of duplicates ($n=2$), see also (Schurch et al. 2016). We however agree that the statistical power in this case is indeed comparably low and we have therefore ensured that our conclusions in the text are appropriately toned.

2). The authors did not report details in their material and method of how many million reads they sequenced per sample, the mapping rates, and the Pearson correlation coefficient value among replicates for data quality/reproducibility assessment.

As requested, we now added the million reads and mapping rates in Table S1. Please note that we had already provided the Euclidean distance obtained in DESeq2 in Supp. Fig. S3A

(now Fig. S3C, Anopheles RNA-seq), which showed the similarity and clustering of our replicates for quality / reproducibility. We have now additionally added the Euclidian distances for the Drosophila dataset in Fig. S3. For the reviewer, we also generated a heatmap of the Pearson correlation coefficients, which confirm the result of Euclidian distance and excellent quality of our datasets. These plots can be also added to the manuscript, if required by the reviewer, although they are to some degree redundant with the Euclidian distances.

Reviewer Figure: Pearson correlation coefficient for RNA-seq performed on *msl-2* null mutant Drosophila (left) and Anopheles embryos (right). Correlation values and heatmaps were obtained with deepTools multiBigWigSummary and plotCorrelation functions. Null mutant and control Drosophila embryos cluster together and show high correlation values. For Anopheles, major clustering is obtained for the different developmental time-points indicative of the entirely different transcriptional programmes along embryogenesis / development. The early time-point datasets (6h-7h) cluster according to replicates corroborating that there are only minor differences between control (WT) and gMSL2 condition. At the 15h time-point clustering of control versus gMSL2 is obtained.

It is unclear when does dosage compensation occur in Agam embryos. In *msl-2* KO embryos, it is difficult to tell when do they arrest in the development stages. Do male/female embryos have different development rate, and do they all passed ZGA and whether X dosage compensation is initiated? Have you done any co-IP to see if MSL complex has formed yet? Maybe some immunostainings of *msl* proteins in early embryos?

Anopheles ZGA occurs approximately after 3h of development (Goltsev et al. 2007, 2004). Since we do not detect any major changes in *msl-2* KO embryos at 6-7h of development (RNA-seq) we conclude that ZGA and early developmental transcriptomes are unaffected. Dosage compensation is in place at the larval and complete at later stages, e.g. pupae and adults (Rose et al. 2016; Papa et al. 2017). We have attempted to detect MSL proteins by Western blot, which would be a requirement to perform co-IPs. Unfortunately, our Drosophila antibodies are not compatible with mosquito (this is why we had performed the polysome profiling experiment in Fig. 1) and this also makes immunofluorescence not possible.

Generating Anopheles-specific antibodies was not feasible in the timeframe of this revision. However, we have now performed other analyses of *msl-2* KO embryos to further pinpoint the developmental arrest (see below). We have also amended our text to clarify these helpful comments raised by the reviewer.

The observation of H4K16ac distributed equally across chromosomes reminds me of another dipteran *Sciara Ocellaris* and its MSL complex equally distributed on X chromosome and autosomes (Ruiz et al 2000). This could be a conserved mechanism within Dipteran Nematocera suborder species? The author should consider discussing this in their discussions.

This is a great point, which we have now added to the discussion. It is interesting that *Sciaria* has a really peculiar sex chromosome system and undergoes X chromosome elimination, so probably is not directly comparable to *Anopheles* and *Drosophila*. Future studies will be required to resolve this really exciting question from a mechanistic point-of-view.

Specific comments:

Fig. 1B. Do these synteny blocks pairs generated over genomic sequences or just for ortholog genes?

Synteny blocks in Fig. 1B are generated over genomic regions of at least 100 Kb (individual 1:1 orthologue genes are shown in Fig. 1C). The Figure legend has been clarified.

Fig 1E and Fig. S1 B, C, D. The qPCR and mRNA-seq data presented in these 4 plots did not provide any statistical test to support their claim on sex-biased differential expression. For example, the authors state "we found a modestly biased expression of *msl2*, *msl3*, *mod* and *mle* in males", and "In RNA-seq dataset, Agam *msls* we all classified as not sex-biased." To just visually access (without any p-value from test), these *msl* genes show higher expression in males in qPCR data but a higher expression in females in mRNA-seq data. Have you done any statistical test to support your statement of biased/not biased expression?

We thank the reviewer for the suggestion. We have added the statistical tests for our experimental qPCR data (Fig. 1E, Fig. S1B and Table S1 for details). For the Fig S1C, we have added the *p*.adj and classification provided by the authors of this published RNA-seq dataset (from Supplemental_Table_S1 in (Papa et al. 2017)). The values in Fig. S1D are obtained from flybase and unfortunately do not contain replicates, hence we could not perform the statistical testing.

What is the normalization used in RNA-seq expression value, are they normalized by read length and library depth?

Our data was normalized with DESeq2, which uses an unnormalized featureCounts file as input (Love et al. 2014, 2021). DESeq2 then normalizes the counts with a linear model, which is described in detail in the aforementioned references (section “The DESeq2 model”). In brief, DESeq2 uses a generalized linear model, where counts K_{ij} for gene i , sample j are modeled using a negative binomial distribution with fitted mean, μ_{ij} and a gene-specific dispersion parameter α_i . The fitted mean is composed of a sample-specific size factor s_j and a parameter q_{ij} proportional to the expected true concentration of fragments for sample j . The coefficients β_i give the log2 fold changes for gene i for each column of the model matrix X . We have now expanded the respective methods section for clarification.

I also will not use the word "validated", first these data are generated from different tissues, and the results are not consistent.

We have changed this wording to “also analyzed”.

Moreover, there is no qPCR data on these msl genes in Dmel as a comparison, does it also show such male-biased in these msl genes?

We have now added the qPCR validation for *Drosophila melanogaster* (New Fig. S1E). As expected, we only detect a strong male bias for *roX2* and a mild bias for *mle*. Since *msl-2* regulation occurs by translational regulation, its RNA levels, as expected and broadly documented in the literature, are equal between males and females (Beckmann et al. 2005; Gebauer et al. 1999).

Fig. 1F and Fig S1E, are these sexed embryos? If they are pooled sex-mixed embryos, the observation of msl genes mRNA drop along development does not mean much regarding to male X dosage compensations.

These are mixed-sex data, that are publicly available. Our point was to look at the expression of *msls* along developmental progression, rather than to comment on the sex differences and/or dosage compensation. We have now rephrased this section to make our intention more clear.

Does Agam have orthologous to roX RNAs?

No. We had already commented on roX RNAs on p. 4 of our text:

“note that the high evolutionary turnover of non-coding RNAs hinders the computational identification of putative orthologues of Dmel roX1 and roX2 (Quinn et al. 2016)”

Fig 2B and Fig S2A. When do they stop development in *msl2*-KO embryos? Do they pass Zygotic genome activation? The embryos seem dead at 30h, do you know if they still viable in 40h and 46h? Is there any way to assess this?

The embryos certainly pass ZGA (around 3h after egg laying) and progress normally to at least 6h-7h of development. For example, in our 6-7h RNA-seq dataset we hardly score any gene expression changes, although *msl-2* is clearly knock-out, indicating that up to this stage, development progresses normally as in the wild-type. At the 15h time-point, we then score many differentially expressed genes.

To complement this data, we have now performed a TUNEL staining in wild-type and knock-out embryos across several time-points (6-7h, 15h, 24h) to detect dead cells in the embryo. This new data (Fig 2C and Fig. S2) supports the RNA-seq and shows that dying cells in the embryos can be detected from the 15h time-point onwards, but not in the early time point (6h-7h). The amount of TUNEL positive nuclei becomes then more severe after 24 hours of development.

Fig. 2E. What stage are these *msl2*-KO embryos? Do you have some immunostainings data of *msl* proteins in early embryos?

As already mentioned above, we have tried immunostainings, but our *Drosophila* antibodies are unfortunately not compatible/specific in *Anopheles*. We have now performed a TUNEL staining instead (see previous point) to further characterize the KO embryos.

Fig. 2F. Here is only for females, any data/image on F1 male testis atrophy?

Upon revision, we have analyzed the male testis in the F1 progeny and now provide some Brightfield pictures in Fig. S2. In *msl-2* mutants, the testis was present and appeared structurally normal. Unfortunately, these tissues were very fragile, hence we were unable to further characterize the tissue and germline.

Fig. 3 and Fig 3s. What stage are these *Drosophila* MSL2-knockdown embryos? Are they comparable to 6-7hr *Agam* embryos regarding the developmental stage? One of the reasons you don't have as many DEGs in *Agam* dataset could be that you have fewer replicates (1 or 2) and from mixed time points compared to in *Drosophila* you have four replicates. What is the Pearson correlation value among *Agam* embryos? Fig 3D, with only 2 replicates in 15hr *Agam* embryos compare to 4 replicates in *drosophila*, I won't be too surprised that you get a smaller number of DEGs.

We thank the reviewer for raising this point. The *Drosophila* embryos represent an overnight collection of heterozygous versus homozygous *msl-2^d* null alleles. They are therefore rather comparable to the 15h and 24h time-points in *Anopheles* (note that *Anopheles*

embryogenesis takes about double the time of *Drosophila*). The reviewer is right - the *Drosophila* dataset with $n=4$ replicates has more statistical power compared to the *Anopheles* dataset ($n=3$ for 6-7h and $n=2$ for 15h) with regards to the absolute number of called DEGs. Nevertheless, even by adding one more replicate the 6-7h time-point dataset in *Anopheles* (only 10 DEG) would not reach the order of magnitude of the *Drosophila* one (more than 1000 DEG). The reason is delineated in (Schurch et al. 2016): To double the amount of scored DEGs, it would be necessary to sequence 10 replicates instead of 3 (i.e. 10 replicates to be able to score 20 DEGs in 6-7h *Anopheles msl-2* KO embryos).

For the 15h time-point, while adding more replicates would definitely reveal more DEGs, this would also not change the major take home-message - X is not different from autosome. Furthermore, in Fig. 3C/3F, we analyze all genes, irrespective of whether they are scored as DE or not.

Regarding the Pearson correlation, see above.

We have now again ensured our text does not make any strong statements about the absolute number of DEG between *Drosophila* and *Anopheles*.

I also don't understand this sentence "the majority of downregulated targets (523 of 566) were affected by at least 75% or more". Where is this 523 from? And what is 75% relative to?

We apologize, if this was not clear. This statement concerns the fold-change observed for the affected downregulated MSL2 targets. For a dosage compensation-like effect, one would expect a fold change of 2, i.e. the targets are downregulated to 50% of the reference level. What we find in *Anopheles* is the following: From the 566 DE down genes, 523 genes are at 25% of the control or lower (e.g. control level is 100%, in the KO the gene drops to 25% or lower). Hence, they are affected by 75% or more. This indicates that *msl-2* KO in *Anopheles* is not associated with a DC-like, 50%-signature that we clearly find in *Drosophila*. We hope this clarifies our calculations.

Fig. 4 and Fig S4. How do you localize the histone marks to the X chromosome in these 2 figures? Have you done X chromosome DNA-FISH to see if you can have X chromosome territory overlap with these histone marks?

We have not done any DNA-FISH, because the localization of the X chromosome territory in *Drosophila* has been widely observed and validated over decades by many labs in the field. The staining looks indeed very typical and as expected.

See for example the following panel taken from our previously published study ((Valsecchi et al. 2018) that shows the territory in males, but not females, WOR refers to the wild-type white Oregon R).

For *Anopheles*, we did not attempt the co-FISH, since we have a ChIP-seq experiment in Fig. 5, that provides much higher resolution and quantitative information compared to immunofluorescence.

Fig. 4C-D, Do these 2 tissue comparable since the salivary glands have polytene chromosomes? Can you show H4K16ac in *Drosophila* Malpighian as a better comparison? Or staining H4K16ac in Agam - do they have polytene chromosomes? It seems conserved.

Please note that H4K16ac enrichment on the X in *Drosophila* occurs independently of ploidy and tissue. To clarify this point, we have now performed further immunostainings and amended the figure panels (see Fig. 4) so that the tissues are now comparable:

Malpighian tubules (polyploid, see e.g. <https://elifesciences.org/articles/54096>) and salivary glands (polyploid), and gut cells (diploid). For color consistency, we have now also amended the channels.

Fig. 5. Why compare ChIP-seq generated from Agam midgut to Dmel L3 Larvae? Shouldn't Larvae in both groups be a better comparison?

We have performed the experiment from *Anopheles* adults, because at this stage, the sex (i.e. males vs. females) can be visually distinguished. Since we had prepared the *Anopheles* chromatin by digestion with Micrococcal nuclease (MNase), which is different from sonication-based methods, our intention was to report a fair comparison between *Anopheles* and *Drosophila* ChIP-seq. We have therefore used a *Drosophila* H4K16ac profile, where MNase was used: Our earlier published profiles from L3 larvae (Valsecchi et al. 2018; note that besides the usage of MNase, the experiments were performed with essentially identical conditions, the very same antibody, library preparation, sequencing pipeline, etc. Nevertheless, we have now added a screenshot that compares *Drosophila* H4K16ac data from different stages/cells (embryo, S2 cells, adults, larvae) revealing that H4K16ac shows a similar profile among all conditions (supplementary Fig. S5C). Indeed, dosage compensation - once

set up - is broadly associated with H4K16ac in various different tissues and cells types. Hence, we think that the comparison shown in the main figure is valid.

Fig. S6. Is Student's t-test the right one? A non-parametric test might be better here for distribution.

The density plots including statistics and p -values are directly obtained from the ShinyGO website <http://bioinformatics.sdstate.edu/go/> (Ge et al. 2020). There is no option for another statistical method on the website and unfortunately, the raw density distribution cannot be downloaded. Hence, we can only report the t-test result obtained from the website.

Fig. 6A, Did this statistical test between DE genes and expected as 2 groups, or it can be done pair-wisely in each exon numbers. If so you can put an asterisk on top of exon number 4 to 9.

The p -value was also obtained from ShinyGO, where it is calculated with the `chisq.test` function in R. Pearson's chi-squared test is performed of the null hypothesis that the joint distribution of the cell counts in a 2-dimensional contingency table is the product of the row and column marginals. It is comparing 2 groups and reports a single p -value.

Fig. 6G. Need to label the sample group on the left of the genomic track.

This has been fixed.

RTR References

- Alekseyenko AA, Peng S, Larschan E, Gorchakov AA, Lee O-K, Kharchenko P, McGrath SD, Wang CI, Mardis ER, Park PJ, et al. 2008. A sequence motif within chromatin entry sites directs MSL establishment on the Drosophila X chromosome. *Cell* **134**: 599–609.
- Beckmann K, Grskovic M, Gebauer F, Hentze MW. 2005. A dual inhibitory mechanism restricts msl-2 mRNA translation for dosage compensation in Drosophila. *Cell* **122**: 529–540.
- Gebauer F, Corona DF, Preiss T, Becker PB, Hentze MW. 1999. Translational control of dosage compensation in Drosophila by Sex-lethal: cooperative silencing via the 5' and 3' UTRs of msl-2 mRNA is independent of the poly(A) tail. *EMBO J* **18**: 6146–6154.
- Ge SX, Jung D, Yao R. 2020. ShinyGO: a graphical gene-set enrichment tool for animals and plants. *Bioinformatics* **36**: 2628–2629.
- Goltsev Y, Fuse N, Frasch M, Zinzen RP, Lanzaro G, Levine M. 2007. Evolution of the dorsal-ventral patterning network in the mosquito, *Anopheles gambiae*. *Development* **134**: 2415–2424.
- Goltsev Y, Hsiung W, Lanzaro G, Levine M. 2004. Different combinations of gap repressors for common stripes in *Anopheles* and *Drosophila* embryos. *Dev Biol* **275**: 435–446.

- Kelley RL, Solovyeva I, Lyman LM, Richman R, Solovyev V, Kuroda MI. 1995. Expression of msl-2 causes assembly of dosage compensation regulators on the X chromosomes and female lethality in *Drosophila*. *Cell* **81**: 867–877.
- Love MI, Anders S, Huber W. 2021. Analyzing RNA-seq data with DESeq2. <http://bioconductor.org/packages/devel/bioc/vignettes/DESeq2/inst/doc/DESeq2.html> (Accessed February 9, 2021).
- Love MI, Huber W, Anders S. 2014. Moderated estimation of fold change and dispersion for RNA-seq data with DESeq2. *Genome Biol* **15**: 550.
- Papa F, Windbichler N, Waterhouse RM, Cagnetti A, D'Amato R, Persampieri T, Lawniczak MKN, Nolan T, Papathanos PA. 2017. Rapid evolution of female-biased genes among four species of *Anopheles malaria* mosquitoes. *Genome Res* **27**: 1536–1548.
- Park Y, Kelley RL, Oh H, Kuroda MI, Meller VH. 2002. Extent of chromatin spreading determined by roX RNA recruitment of MSL proteins. *Science* **298**: 1620–1623.
- Quinn JJ, Zhang QC, Georgiev P, Ilik IA, Akhtar A, Chang HY. 2016. Rapid evolutionary turnover underlies conserved lncRNA-genome interactions. *Genes Dev* **30**: 191–207.
- Ramírez F, Lingg T, Toscano S, Lam KC, Georgiev P, Chung H-R, Lajoie BR, de Wit E, Zhan Y, de Laat W, et al. 2015. High-Affinity Sites Form an Interaction Network to Facilitate Spreading of the MSL Complex across the X Chromosome in *Drosophila*. *Mol Cell* **60**: 146–162.
- Rose G, Krzywinska E, Kim J, Revuelta L, Ferretti L, Krzywinski J. 2016. Dosage Compensation in the African Malaria Mosquito *Anopheles gambiae*. *Genome Biol Evol* **8**: 411–425.
- Schurch NJ, Schofield P, Gierliński M, Cole C, Sherstnev A, Singh V, Wrobel N, Gharbi K, Simpson GG, Owen-Hughes T, et al. 2016. How many biological replicates are needed in an RNA-seq experiment and which differential expression tool should you use? *RNA* **22**: 839–851.
- Sun L, Fernandez HR, Donohue RC, Li J, Cheng J, Birchler JA. 2013. Male-specific lethal complex in *Drosophila* counteracts histone acetylation and does not mediate dosage compensation. *Proc Natl Acad Sci U S A* **110**: E808–17.
- Sun X, Birchler JA. 2009. Interaction study of the male specific lethal (MSL) complex and trans-acting dosage effects in metafemales of *Drosophila melanogaster*. *Cytogenet Genome Res* **124**: 298–311.
- Valsecchi CIK, Basilicata MF, Georgiev P, Gaub A, Seyffert J, Kulkarni T, Panhale A, Semplicio G, Manjunath V, Holz H, et al. 2021. RNA nucleation by MSL2 induces selective X chromosome compartmentalization. *Nature* **589**: 137–142.
- Valsecchi CIK, Basilicata MF, Semplicio G, Georgiev P, Gutierrez NM, Akhtar A. 2018. Facultative dosage compensation of developmental genes on autosomes in *Drosophila* and mouse embryonic stem cells. *Nat Commun* **9**: 3626.

June 18, 2021

RE: Life Science Alliance Manuscript #LSA-2020-00996-TR

Dr. Asifa Akhtar
Max Planck Institute of Immunobiology and Epigenetics
Stübeweg 51
Freiburg 79108
Germany

Dear Dr. Akhtar,

Thank you for submitting your revised manuscript entitled "Distinct mechanisms mediate X chromosome dosage compensation in *Anopheles* and *Drosophila*". We would be happy to publish your paper in Life Science Alliance pending final revisions necessary to meet our formatting guidelines.

- please add ORCID ID for the corresponding author-you should have received instructions on how to do so
- please add callouts for Figures 6F, S1D to your main manuscript text

Figure checks:

- scale bars in Figure S2B are hard to read
- Please add scale bars for Figures 2B, S2A, D

A. FINAL FILES:

B. MANUSCRIPT ORGANIZATION AND FORMATTING:

Sincerely,

Reviewer #1 (Comments to the Authors (Required)):

I appreciate the fact that the authors made an effort to address the normalization issue. I am not sure that the failed depletion of rRNA is the best normalization. One might wonder if there is any evidence that the failed depletion is consistent across samples. What would be the very best normalization is to perform a transcriptome size measurement, which RNA-seq alone cannot do. Loven et al (Loven, J., Orlando, D.A., Sigova, A.A., Lin, C.Y., Rahl, P.B., Burge, C.B., Levens, D.L., Lee, T.I. and Young, R.A. (2012). Revisiting global gene expression analysis. *Cell* 151: 476-482.) pointed out that the whole transcriptome size can increase and what appears to be a down regulation is just not upregulated as much. What the authors report with a slight autosomal increase and a presumed down regulation of the X is exactly the type of result that Loven et al would predict from a transcriptome size increase. Indeed, if one looks at the data in Belote and Lucchesi (Belote, J.M., and Lucchesi, J.C. (1980). Control of X chromosome transcription by the maleless gene in *Drosophila*. *Nature* 285: 573-575.) and normalizes the nascent RNA grains to the DNA grains, one will find that the X is not reduced and the autosomes are increased. Now, it is the case that monosomy can have positive effects across the genome but that does not seem to be the case and it is not supported by the Belote and Lucchesi data (nor does it fit the claims of the authors). I am not sure that polymerase III transcribed genes is a good idea for normalization either since tRNAs are dosage compensated (Birchler et al., 1982, *Genetics* 102:525-537).

The inverse effect is noticeable in the following *Drosophila* publications, so it probably should not be discounted in considering how dosage compensation operates.

- Birchler JA: Expression of cis-regulatory mutations of the white locus in metafemales of *Drosophila melanogaster*. *Genet Res* 59:11-18 (1992).
- Birchler JA, Hiebert JC, Krietzman M: Gene expression in adult metafemales of *Drosophila melanogaster*. *Genetics* 122: 869-879 (1989).
- Birchler JA, Hiebert JC, Paigen K: Analysis of autosomal dosage compensation involving the alcohol dehydrogenase locus in *Drosophila melanogaster*. *Genetics* 124: 677-686 (1990).
- Devlin RH, Holm DG, Grigliatti TA: Autosomal dosage compensation in *Drosophila melanogaster* strains trisomic for the left arm of chromosome 2. *Proc. Natl. Acad. Sci. USA* 79 1200-1204 (1982).
- Devlin RH, Grigliatti TA, Holm DG: Dosage compensation is transcriptionally regulated in autosomal trisomics of *Drosophila*. *Chromosoma* 91: 65-73 (1984).
- Devlin RH, Holm DG, Grigliatti TA: The influence of whole-arm trisomy on gene expression in *Drosophila*. *Genetics* 118: 87-101 (1988).
- Hall J, Kankel DR: Genetics of acetylcholinesterase in *D. melanogaster*. *Genetics* 83:517-533 (1976).
- Hiebert JC, Birchler JA: Effects of the maleless mutation on X and autosomal gene expression in *Drosophila melanogaster*. *Genetics* 136:913-926 (1994).
- Hodgetts RB: Response of DOPA decarboxylase activity variations in gene dosage in *Drosophila*: A possible location of the structural gene. *Genetics* 79:45-54 (1975).
- Muller HJ: Evidence for the precision of genetic adaptation. *The Harvey Lectures, Series XLIII*. Chas. C. Thomas, Springfield, IL. Pp. 165-229 (1950).
- O'Brien SJ, Gethmann RC: Segmental aneuploidy as a probe for structural genes in *Drosophila*: mitochondrial membrane enzymes. *Genetics* 75: 155-167 (1973).
- Oliver MJ, Huber RE, Williamson JW: Genetic and biochemical aspects of trehalase from *Drosophila melanogaster*. *Biochem Genetics* 16:927-940 (1978).
- Pal-Bhadra M, Bhadra U, Birchler JA: Gene expression analysis of the function of the male-specific lethal complex in *Drosophila*. *Genetics* 169:2061-2074 (2005).
- Pipkin SB, Chakrabarty PK, Bremner TA: Location and regulation of *Drosophila* fumarase. *J Heredity* 68:245-252 (1977).
- Rabinow L, Nguyen-Huynh AT, Birchler JA: A trans-acting regulatory gene that inversely affects the

expression of the white, brown and scarlet loci in *Drosophila*. *Genetics* 129: 463-480 (1991).
Rawls JM, Lucchesi JC: Regulation of enzyme activities in *Drosophila*. I. The detection of regulatory loci by gene dosage responses. *Genet Res* 24:59-72 (1974).
Sabl JF, Birchler JA: Dosage dependent modifiers of white alleles in *Drosophila melanogaster*. *Genet Res* 62: 15-22 (1993).
Sun L, Fernandez HR, Donohue RC, Li J, Cheng J, Birchler JA: Male-specific lethal complex in *Drosophila* counteracts histone acetylation and does not mediate dosage compensation. *Proc Natl Acad Sci USA* 110: E808-817 (2013).
Sun L, Johnson AF, Donohue RC, Li J, Cheng J, Birchler JA: Dosage compensation and inverse effects in triple X metafemales of *Drosophila*. *Proc. Natl. Acad. Sci. U.S.A.* 110: 7383-7388 (2013).
Sun L, Johnson AF, Li J, Lambdin AS, Cheng J, Birchler JA: Differential effect of aneuploidy on the X chromosome and genes with sex-biased expression in *Drosophila*. *Proc. Natl. Acad. Sci. U.S.A.* 110: 16514-16519 (2013).

Our lab has beat to death targeting. When we target MOF, there is an increase in expression of the target but in all circumstances in which the whole MSL complex is present or is targeted itself (individual genes or whole chromosomes), there is no discernible effects including phenotypically, which does not suffer from any normalization discussions. If there were an effect, so be it, but we just can't find that.

Having stated all of the above, the authors are correct that what is occurring in mosquito and fly is different. They are also correct that this manuscript is not the place to try to resolve all of this.

Reviewer #2 (Comments to the Authors (Required)):

The manuscript has been greatly improved after revision. All my concerns have been addressed and I have no other comments to make.

June 28, 2021

RE: Life Science Alliance Manuscript #LSA-2020-00996-TRR

Dr. Asifa Akhtar
Max Planck Institute of Immunobiology and Epigenetics
Stübeweg 51
Freiburg 79108
Germany

Dear Dr. Akhtar,

Thank you for submitting your Research Article entitled "Distinct mechanisms mediate X chromosome dosage compensation in Anopheles and Drosophila". It is a pleasure to let you know that your manuscript is now accepted for publication in Life Science Alliance. Congratulations on this interesting work.

DISTRIBUTION OF MATERIALS:

Again, congratulations on a very nice paper. I hope you found the review process to be constructive and are pleased with how the manuscript was handled editorially. We look forward to future exciting submissions from your lab.

Sincerely,
